# Comparing frequency of booster vaccination to prevent severe COVID-19 by risk group in the United States

Hailey J. Park[1], Gregg S. Gonsalves [2], Sophia T. Tan[1], J. Daniel Kelly [3,4,5,6,7], George W. Rutherford[3,7], Robert M. Wachter [6], Robert Schechter[8], A. David Paltiel[9] & Nathan C. Lo [1]✉

There is a public health need to understand how different frequencies of COVID-19 booster vaccines may mitigate the risk of severe COVID-19, while accounting for waning of protection and differential risk by age and immune status. By analyzing United States COVID-19 surveillance and seroprevalence data in a microsimulation model, here we show that more frequent COVID-19 booster vaccination (every 6–12 months) in older age groups and the immunocompromised population would effectively reduce the burden of severe COVID-19, while frequent boosters in the younger population may only provide modest benefit against severe disease. In persons 75+ years, the model estimated that annual boosters would reduce absolute annual risk of severe COVID-19 by 199 (uncertainty interval: 183–232) cases per 100,000 persons, compared to a one-time booster vaccination. In contrast, for persons 18–49 years, the model estimated that annual boosters would reduce this risk by 14 (10–19) cases per 100,000 persons. Those with prior infection had lower benefit of more frequent boosting, and immunocompromised persons had larger benefit. Scenarios with emerging variants with immune evasion increased the benefit of more frequent variant-targeted boosters. This study underscores the benefit of considering key risk factors to inform frequency of COVID-19 booster vaccines in public health guidance and ensuring at least annual boosters in high-risk populations.

Both COVID-19 vaccination and natural infection from SARS-CoV-2 generate protection against future risk of COVID-19; however, this protection wanes over time, in part due to new variants[1–6]. While waning protection from vaccination and natural infection against SARS-CoV-2 infection is well-documented[6], recent analyses have also found some waning of protection against severe COVID-19 (defined as hospitalization or death)[1–4,7]. Studies further suggest that additional booster vaccine doses or natural infection can restore the level of protection despite this prior decline[1,3,8]. A key question remains: what is the comparative effectiveness of different frequencies of COVID-19

[1]Division of Infectious Diseases and Geographic Medicine, Department of Medicine, Stanford University, Stanford, CA, USA. [2]Department of Epidemiology of Microbial Diseases and Public Health Modeling Unit, Yale School of Public Health, New Haven, CT, USA. [3]Department of Epidemiology and Biostatistics, University of California, San Francisco, San Francisco, CA, USA. [4]San Francisco Veterans Affairs Medical Center, San Francisco, CA, USA. [5]F.I. Proctor Foundation, University of California, San Francisco, San Francisco, CA, USA. [6]Department of Medicine, University of California, San Francisco, San Francisco, CA, USA. [7]Institute for Global Health Sciences, University of California, San Francisco, San Francisco, CA, USA. [8]California Department of Public Health, Richmond, CA, USA. [9]Department of Health Policy and Management and Public Health Modeling Unit, Yale School of Public Health, New Haven, CT, USA. ✉e-mail: Nathan.Lo@stanford.edu

booster vaccination in key risk groups to offset waning of protection against severe disease?

While there was considerable study of vaccine prioritization during the introduction of the COVID-19 vaccine[9–13], there is limited evidence to guide decisions on the timing of COVID-19 booster vaccination to prevent severe COVID-19. The risk for severe COVID-19 is complex and person-specific. Considerations to determine the frequency of COVID-19 booster vaccination (e.g., monovalent vaccines targeting one variant such as Omicron XBB.1.5, or bivalent vaccines targeting more than one variant, such as the ancestral strain and Omicron subvariants BA.4/5) for an individual include: i) baseline risk for severe COVID-19 given infection, correlated with increased age and presence of immunocompromising conditions; ii) vaccination history, including number of doses and time since last vaccination; iii) previous SARS-CoV-2 infection(s), including time since last infection and variant; and iv) overall risk of infection driven by levels of SARS-CoV-2 transmission in the community. Given heterogeneity in risk of severe COVID-19 within the population, the comparative effectiveness of different frequencies of COVID-19 booster vaccination may vary based on key risk factors.

While limiting SARS-CoV-2 community transmission and providing access to antiviral treatment for COVID-19 is important from a public health perspective, here we focus on the impact of the timing of booster vaccination in different age groups and the immunocompromised population to prevent severe disease. Using a microsimulation model, which is a common public health modeling approach that allows the simulation of individual people with unique characteristics[10,14,15], we model SARS-CoV-2 infection and severe COVID-19 to compare the impact of various timings of COVID-19 booster vaccination in different risk groups. The aim of this study is to inform guidance for the frequency of COVID-19 booster vaccination in the United States.

## Results

### Primary model results

We analyzed detailed COVID-19 surveillance data and seroprevalence estimates from the US Centers for Disease Control and Prevention and assessed vaccine strategies beginning in September 2022. We developed and calibrated a microsimulation model of severe COVID-19 to a simulated population composed of vaccinated persons (with a primary series with BNT162b2 or mRNA-1273, plus at least one monovalent mRNA booster) in four age groups: 18-49 years, 50-64 years, 65-74 years, and 75+ years, and an immunocompetent and immunocompromised population (mild, moderate/severe). Model inputs included person-level vaccination history and probability of prior infection, which informed the waning of protection since last vaccine dose or natural infection.

Over a two-year period, the primary model estimated that more frequent COVID-19 booster vaccination in older age groups would have larger absolute reductions in severe COVID-19 risk (Table 1). In a hypothetical cohort of persons 18-49 years old who received a one-time booster vaccination, the model estimated an annual risk of 98 (uncertainty interval (UI): 85-125) severe cases per 100,000 persons. The model estimated annual booster vaccination would reduce relative annual risk of severe COVID-19 by 14% and absolute risk by 14 (UI: 10-19) cases per 100,000 persons (number of persons needed to treat (NNT) 3534; over 2-year period), compared to a base case of a one-time booster vaccination. The model estimated that semiannual (every 6 months) booster vaccination would reduce relative annual risk by 27% and absolute risk by 26 (UI: 21-35) cases per 100,000 persons (NNT 1916), compared to a one-time booster vaccination. In the annual and semiannual vaccine strategy, we estimated that 48% and 46% of the averted severe COVID-19 cases occurred in persons without prior documented COVID-19 infection at the start of the simulation, respectively.

In contrast, in a hypothetical cohort of persons 75+ years old who received a one-time booster vaccination, the model estimated an annual risk of 1398 (UI: 1332–1501) severe cases per 100,000. The model estimated annual booster vaccination would reduce relative annual risk of severe COVID-19 by 14% and absolute risk by 199 (UI: 183–232) cases per 100,000 persons (NNT 251), compared to a base case of a one-time booster vaccination. The model estimated that semiannual booster vaccination would reduce relative annual risk by 26% and absolute risk by 368 (UI: 341–415) cases per 100,000 persons (NNT 136), compared to a one-time booster vaccination. In the annual and semiannual vaccine strategies, we estimated that 82% and 82% of the averted severe COVID-19 cases occurred in persons without prior documented COVID-19 infection.

In a hypothetical cohort of mild and moderate/severe immunocompromised persons of all age groups who received a one-time booster vaccination, the model estimated an age-weighted annual risk of 1290 (UI: 1205–1403) and 1367 (UI: 1266–1503) cases per 100,000 persons, respectively. For mild immunocompromised persons, annual and semiannual booster vaccination reduced absolute annual risk by 110 (UI: 58–143) and 195 (UI: 111–252) cases per 100,000 persons respectively, compared to a one-time booster vaccination. For moderate/severe immunocompromised persons, annual and semiannual booster vaccination reduced absolute annual risk by 184 (UI: 164–203) and 310 (UI: 276–342) cases per 100,000 persons respectively, compared to a one-time booster vaccination. Full age-specific estimates for the immunocompromised population are available in Supplementary Tables S10-S11.

Full reporting of results, including for persons 50–64 years and persons 65–74 years, are shown in Table 1. Model validation results demonstrated that model predictions for severe COVID-19 incidence were similar to observed values (Supplementary Table S4). Model predictions on risk of severe COVID-19 without any booster are available in the Appendix (Supplementary Table S12).

### Scenario of novel variants and impact of transmission dynamics

We repeated the primary analysis under different scenarios with emergence of novel variants with immune evasion (summarized in Fig. 1A). Scenarios simulating novel variants with immune evasion increased overall number of severe COVID-19 cases, although the overall impact of more frequent booster vaccines by risk group was similar; uncertainty in this analysis was larger. In those 65–74 years old, annual and semiannual booster vaccination under annual novel variant circulation (scenario 3) would lead to an annual risk reduction of 73 (UI: 56–89) and 134 (UI: 96–154) severe cases per 100,000 persons, respectively, compared to a one-time booster vaccination. Under the primary analysis (without novel variant introduction) this would lead to an annual risk reduction of 78 and 142 severe cases per 100,000 persons. The scenario with a variant-targeted vaccine had larger benefits of more frequent booster vaccines. In persons 65–74 years old, annual and semiannual booster vaccination with a variant-targeted vaccine (scenario 4) would lead to an annual risk reduction of 130 (UI: 118–163) and 233 (UI: 200–272) severe cases per 100,000 persons, respectively, compared to a one-time booster vaccination.

To investigate the impact of indirect effects of vaccination on transmission, we repeated the primary analysis using a dynamic transmission model (Fig. 2). We found that indirect effects were larger with more inclusive frequent booster vaccine strategies, although within the assumed conditions and realistic vaccine uptake, the overall model conclusions were broadly similar to the primary (static) model. In a focused vaccination program in high-risk populations (75+ years and moderate/severe immunocompromised groups) under realistic vaccine coverage assumptions, the dynamic model estimated that annual and semiannual booster vaccination would lead to an annual risk reduction of 209 (UI: 186–258) and 450 (UI: 387–518) severe cases per 100,000 persons in those 75+ years, compared to a one-time

**Table 1 | Number of severe COVID-19 cases, risk, and number needed to treat to avert severe COVID-19 in six risk groups with different frequencies of booster vaccination**

| | Total severe COVID-19 cases | Absolute annual risk of severe COVID-19 (cases per 100,000; UI) | Annual risk reduction of severe COVID-19 | | % Averted severe COVID-19 by infection status[a] | | NNT to avert severe COVID-19 case[b] |
|---|---|---|---|---|---|---|---|
| | | | Absolute risk averted (cases per 100,000) | Relative risk averted (%) | No Prior Infection[a] | Prior Infection[a] | |
| **One-time booster[c]** | | | | | | | |
| 18–49 years | 1954 | 98 (85–125) | -- | -- | -- | -- | -- |
| 50–64 years | 3978 | 199 (185–238) | -- | -- | -- | -- | -- |
| 65–74 years | 10,484 | 524 (499–562) | -- | -- | -- | -- | -- |
| 75+ years | 27,955 | 1398 (1332–1501) | -- | -- | -- | -- | -- |
| Immunocompromised (Mild)[d] | 25,805 | 1290 (1205–1403) | -- | -- | -- | -- | -- |
| Immunocompromised (Moderate/Severe)[d] | 27,343 | 1367 (1266–1503) | -- | -- | -- | -- | -- |
| **Annual booster** | | | | | | | |
| 18–49 years | 1671 | 84 (74–106) | 14 | 14% | 48% | 52% | 3534 |
| 50–64 years | 3424 | 171 (159–202) | 28 | 14% | 68% | 32% | 1806 |
| 65–74 years | 8924 | 446 (425–475) | 78 | 15% | 83% | 17% | 642 |
| 75+ years | 23,966 | 1198 (1144–1272) | 199 | 14% | 82% | 18% | 251 |
| Immunocompromised (Mild)[d] | 23,609 | 1180 (1088–1316) | 110 | 9% | 67% | 33% | 456 |
| Immunocompromised (Moderate/Severe)[d] | 23,669 | 1183 (1091–1307) | 184 | 13% | 50% | 50% | 273 |
| **Semiannual booster (every 6 months)** | | | | | | | |
| 18–49 years | 1432 | 72 (64–90) | 26 | 27% | 46% | 54% | 1916 |
| 50–64 years | 2944 | 147 (136–171) | 52 | 26% | 67% | 33% | 968 |
| 65–74 years | 7645 | 382 (365–404) | 142 | 27% | 83% | 17% | 353 |
| 75+ years | 20,602 | 1030 (988–1088) | 368 | 26% | 82% | 18% | 136 |
| Immunocompromised (Mild)[d] | 21,899 | 1095 (987–1255) | 195 | 15% | 67% | 33% | 257 |
| Immunocompromised (Moderate/Severe)[d] | 21,138 | 1057 (966–1183) | 310 | 23% | 51% | 49% | 162 |

[a]Prior infection status based on start of simulation. Percent averted estimate refers to averted severe COVID-19 cases due to vaccine strategy.

[b]NNT; number needed to treat, which is based on the number of persons (instead of vaccine doses) needing to follow a vaccine schedule to avert one severe COVID-19 case; estimated over 2-year simulation period in population of 1 million persons for each risk group.

[c]One-time booster is the baseline intervention for risk reduction calculations.

[d]Definitions for each immunocompromised status are available in the Methods. We report age-weighted estimates in this Table. Full age-stratified results for the immunocompromised population are available in the Appendix (Supplementary Tables S10, S11).

Scenario with no booster is available in Supplementary Table S12. The uncertainty intervals capture the full range of varied model parameters, while the point estimate uses base case assumptions of model inputs.

booster vaccination (Fig. 2A). In a more inclusive vaccination program (18+ years all groups), the model estimated that annual and semiannual booster vaccination would lead to an annual risk reduction of 257 (UI: 229–295) and 602 (UI: 513–683) severe cases per 100,000 persons in those 75+ years, compared to a one-time booster vaccination (Fig. 2A). Under more optimistic vaccine coverage assumptions, indirect effects were larger (Fig. 2B).

**Sensitivity analysis**

In a sensitivity analysis for the primary model, we found that higher incidence of severe COVID-19 was associated with the largest increase in gains associated with more frequent boosting. For example, in persons 65–74 years old, semiannual booster averted 219 cases per 100,000 persons in the high incidence scenario compared to 142 cases per 100,000 persons in the primary analysis. Additionally, more rapid waning of vaccine-induced protection (pessimistic waning) and higher vaccine effectiveness had larger gains associated with more frequent boosting, although the estimates were overall similar (Fig. 3). We conducted a sensitivity analysis where all persons were assumed to have prior COVID-19 and found similar benefits of more frequent vaccination (Supplemental Table S21). Full results for each sensitivity analysis are available in Supplementary Tables S13–S25.

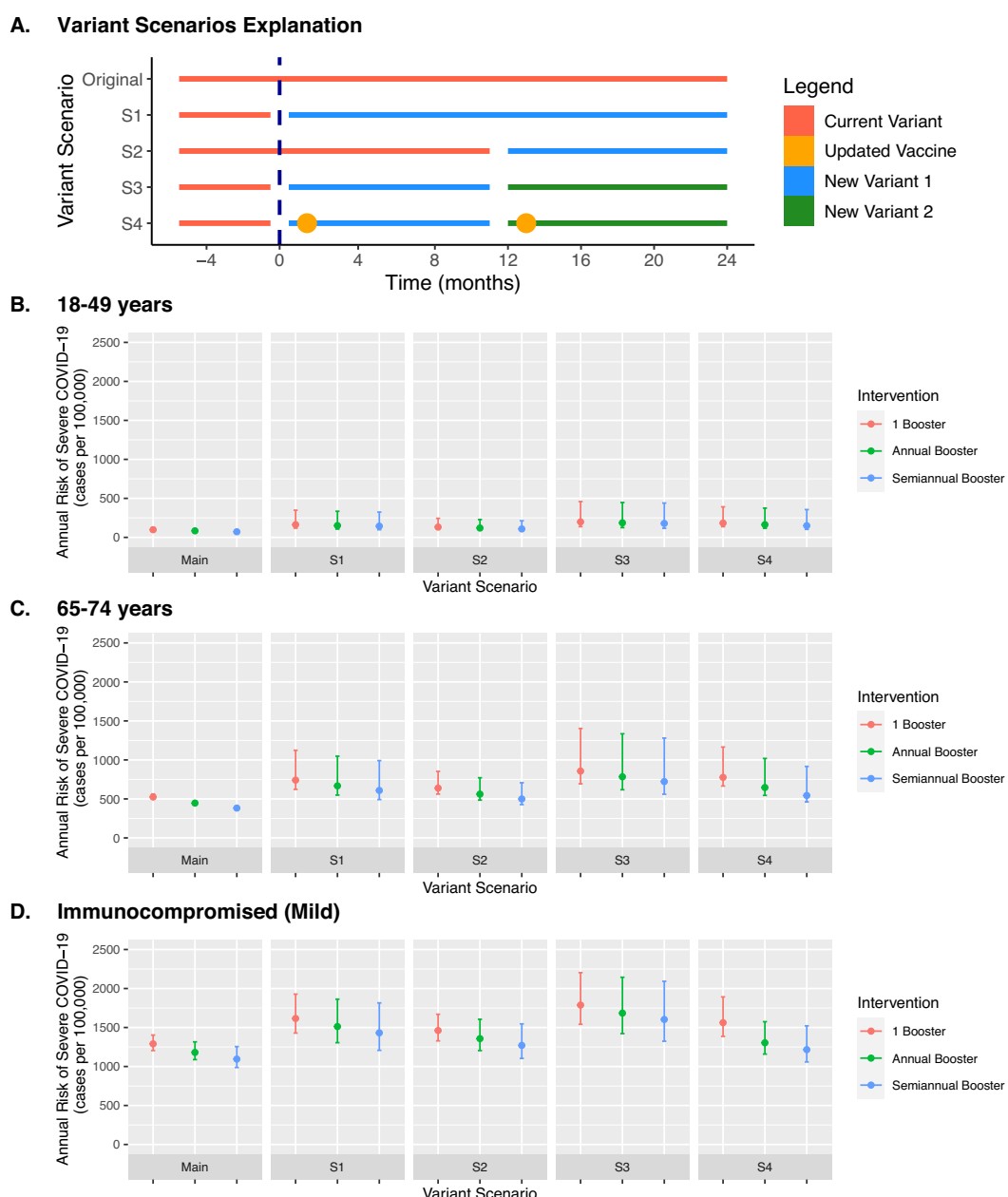

**Fig. 1 | Scenario analysis on emergence of novel SARS-CoV-2 variants comparing severe COVID-19 risk with different frequencies of COVID-19 booster vaccination.** We simulated four scenarios (S1-S4) on emergence of novel variant(s) with 10% reduced susceptibility to the protection generated by prior vaccination and natural infection (**A**), including use of a variant-targeted vaccine (S4). Under each variant scenario analysis, we simulated three frequencies of COVID-19 booster vaccine in each risk group. We display results for three risk groups: 18–49 years (**B**), 65–74 years (**C**), and mild immunocompromised population (**D**). We plotted absolute annual risk of severe COVID-19 over a two-year simulation. The vertical bars represent uncertainty intervals and capture the full range of varied model parameters (*n* = 25 simulations per model parameter set), while the point estimate uses base case assumptions of model inputs. Intervals are designed to demonstrate uncertainty within a single vaccine strategy; comparison between vaccine strategies should use the same assumed baseline conditions. Additional variant scenarios and risk groups available in Supplementary Figs. S12, S13.

## Discussion

To inform guidance on schedules of COVID-19 booster vaccination, this modeling study compared different frequencies of booster vaccination and risk of severe COVID-19 in key age groups and the immunocompromised population. While both COVID-19 vaccination and natural infection generate protection against severe COVID-19, this protection wanes over time, prompting discussion on the optimal timing of booster vaccination[1,2]. We found that more frequent COVID-19 booster vaccination in older populations and those with immuno-compromising conditions at risk for severe COVID-19, along with less frequent booster vaccination in younger low-risk populations, may

efficiently mitigate the burden of severe COVID-19 in the United States. These findings were similar when accounting for indirect effects of vaccines on transmission, although more inclusive vaccination at higher coverage did yield benefits to reduce transmission across all risk groups. We also found that the robustness and durability of hybrid protection lowers the value of repeated boosters, except in cases of variants with immune evasion where prior protection is reduced. Scenarios with emerging variants increased the benefit of more frequent variant-targeted boosters within the assumptions of the model. Our study supports current guidance to provide at least annual boosters for those 65 years and older and/or with

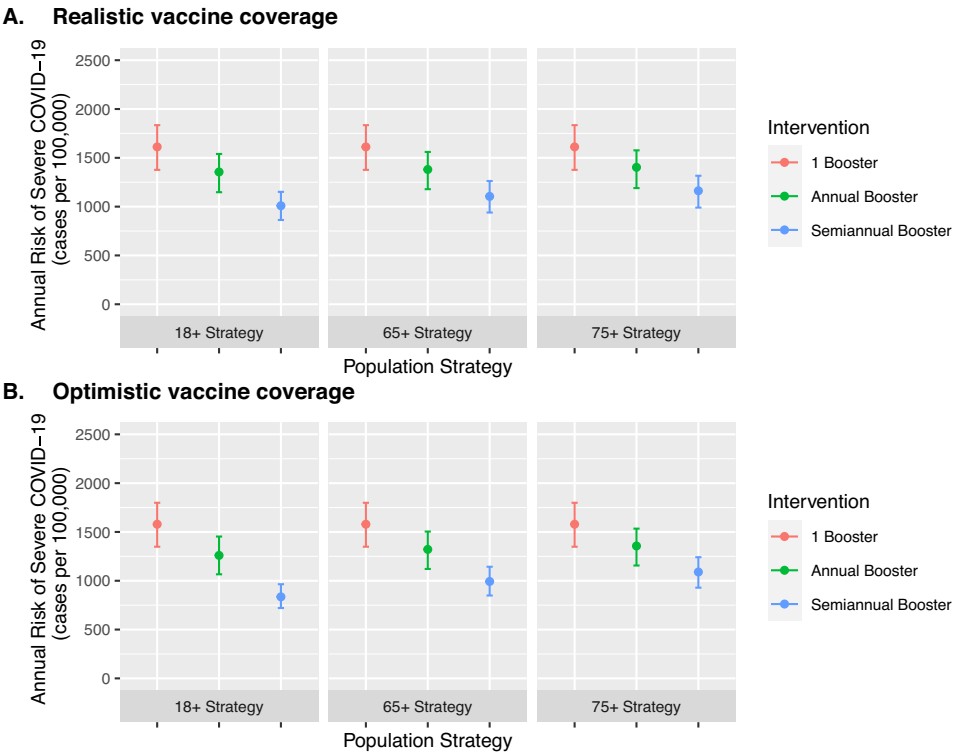

**Fig. 2 | Scenario analysis using a dynamic transmission model to estimate the impact of indirect effects on COVID-19 booster vaccination strategies in the 75 years and older group.** We used a dynamic transmission model to simulate different frequencies of COVID-19 booster vaccination in different eligible risk groups to determine how each vaccine strategy would affect transmission to the highest risk populations (75+ years). We simulated more frequent booster vaccination with varying levels of inclusiveness: (i) 18+ years in all groups (most inclusive); (ii) 65+ years and all immunocompromised groups; and (iii) 75+ years, moderate/severe immunocompromised group (most restrictive). We simulated under realistic vaccine coverage (**A**) and optimistic coverage (**B**) assumptions (Table S7). We assumed a background of one-time booster vaccination at the start of the simulation in adults (18+ years) with age-specific, imperfect vaccine uptake. We plotted absolute annual risk of severe COVID-19 in the 75+ year risk group, to compare the indirect effects of booster vaccination in this high-risk group. Larger indirect effects are expected with more inclusive vaccine strategies. The vertical bars represent uncertainty intervals and capture the full range of varied model parameters ($n = 25$ simulations per model parameter set), while the point estimate uses base case assumptions of model inputs. Intervals are designed to demonstrate uncertainty within a single vaccine strategy; comparison between vaccine strategies should use the same assumed baseline conditions. A full description of the Methods and results for additional risk groups are available in the Appendix.

immunocompromising conditions[16], and illustrates the importance of considering key risk groups when determining guidance for booster timing to reduce risk of severe COVID-19.

The optimal timing and need for COVID-19 booster vaccination will depend on value judgments, what factors are considered, evaluation of the absolute and relative risks of severe COVID-19, and the perspective (individual or population level). Our goal was to provide these estimates to inform vaccine guidance and public health decisions, although these results could be considered for personalized patient and clinician decisions. For interpreting this risk, benefit, and NNT, our results may be contextualized by comparing them to common preventive health measures. For example, common primary care measures have a range of NNT often below 1000, such as influenza vaccination to prevent death, statin for primary prevention of death (NNT 286), and colonoscopy to prevent colon cancer associated death (NNT 445)[17–19]. As an example, an absolute risk threshold such as averting a 1 in 1000 chance of being hospitalized or dying from COVID-19 may be an informative threshold. While relative risks differences were similar across risk groups, the absolute risk differences for severe COVID-19 were much larger in the higher risk groups and a more meaningful measure of risk reduction. Overall, these findings support more frequent booster (e.g., at least annual vaccination) in populations 65 years and older and those with immunocompromising conditions, which broadly align with the absolute risk thresholds and NNT estimates discussed.

Most of the estimated benefit from more frequent booster vaccination occurred in older age groups, the immunocompromised, and those without prior COVID-19, which is consistent with prior literature analyzing vaccine prioritization during the introduction of COVID-19 vaccines[9,10,13]. Less benefit is derived from more frequent booster vaccination in younger, low-risk populations, although there are some indirect effects on transmission (i.e., reducing transmission to higher-risk populations) from doing so as demonstrated in our scenario analysis using a dynamic transmission model. While vaccine guidance on frequency of booster vaccines based on key risk factors defined by age and immunocompromised status would be supported by this study, decisions on vaccine guidelines based on prior COVID-19 disease is likely more challenging. For example, people may misclassify their prior infection status in both directions – by either assuming a prior infection or not recognizing one based on confirmatory testing, and further data is needed to confirm the robustness of hybrid immunity.

The model relied upon available literature estimates and simplifying assumptions on vaccination and the level of protection against severe COVID-19. We assumed that each COVID-19 vaccine booster had comparable vaccine effectiveness. We conservatively assumed that these additional doses did not have higher absolute vaccine effectiveness, but rather restored the maximal protection prior to waning, an assumption that is broadly supported by literature[1,8]. Because of limited long-term data (beyond 12 months of follow up) on the waning

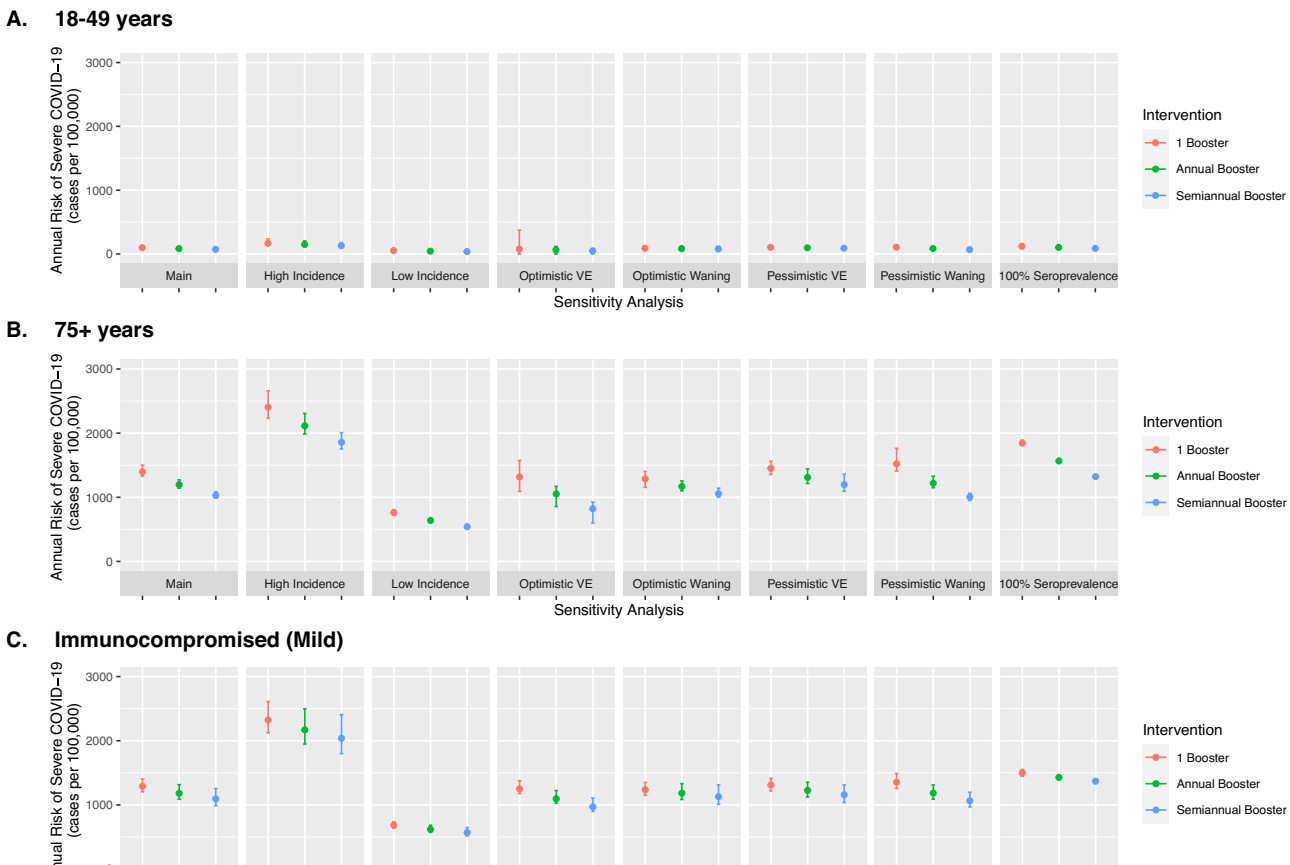

**Fig. 3 | Sensitivity analysis of model parameters for COVID-19 risk and booster vaccination.** This sensitivity analysis tested alternative model parameters and assumptions on overall vaccine-induced protection (optimistic and pessimistic assumptions), waning of vaccine-induced protection (optimistic and pessimistic assumptions), COVID-19 incidence (0.5x lower or 2x higher) and seroprevalence (100% previously infected). For each sensitivity analysis, we re-calibrated the model and simulated three COVID-19 booster vaccine schedules and plotted annual risk of severe COVID-19. We plotted results for three representative risk groups: 18–49 years (**A**), 75+ years (**B**), and the mild immunocompromised population (**C**). The vertical bars represent uncertainty intervals and capture the full range of varied model parameters (*n* = 25 simulations per model parameter set), while the point estimate uses base case assumptions of model inputs. Intervals are designed to demonstrate uncertainty within a single vaccine strategy; comparison between vaccine strategies should use the same assumed baseline conditions. Additional risk groups are available in the Appendix (Supplementary Tables S13–S25).

of vaccine-induced protection, we projected this waning by synthesizing results from multiple published studies and meta-analyses with distributional assumptions[1–3,5] (Appendix, "Technical Appendix"). These data were observational, and therefore may be prone to some biases. The available evidence suggests hybrid immunity provides high and robust protection against severe disease;[1,4,20] however this literature is limited, so additional research to confirm this finding will be important, especially under novel variants. While there is uncertainty in the level of protection and waning over time of vaccine-induced and hybrid immunity, the overall conclusions of the study remained robust under a broad range of assumptions in model initialization explored in sensitivity analyses. This is because different assumptions on level of protection and waning at baseline are offset in the model by changes in force of infection of COVID-19 during the calibration, leading to similar results.

Decisions on frequency of COVID-19 booster vaccination are likely to be influenced by emergence of novel variants and new formulations of vaccine. In this study, we simulated different potential scenarios for emergence of novel variants with immune evasion, although the full range of evolutionary possibilities for variant characteristics (e.g., infectiousness, severity of illness, mechanism of immune evasion) are difficult to capture. We simulated variant scenarios with evasion of

protection generated by vaccine and hybrid immunity and found our overall study findings to be robust, with larger benefit using variant-targeted vaccines. These study findings will be relevant to new COVID-19 vaccine formulations (e.g., Omicron XBB.1.5) if the generated protection against circulating variants is similar to the data and assumptions used in the study.

To evaluate the importance of indirect effects of more frequent booster vaccination on transmission[21–23], we performed a scenario analysis using a dynamic transmission model since the primary analysis did not account for indirect effects. While we found transmission reductions in all groups (including the high-risk groups) due to indirect effects caused by more frequent booster vaccination in more inclusive strategies (e.g., 18+ years in all groups), the overall model conclusions were similar. This is likely since: i) booster vaccines yield indirect protection, but these effects are relatively short-lived and modest;[21–23] ii) vaccine booster uptake is overall low within the population based on current coverage estimates; iii) a substantial proportion of the population remains unvaccinated or under vaccinated. Therefore, under reasonable assumptions, the indirect effects of booster vaccination are unlikely to change the study conclusions that consider direct protection alone. Furthermore, vaccine guidance for boosters may be more likely to primarily consider direct protection from vaccination. This

reinforces the validity of using a static model, which relies on fewer assumptions, for the primary analysis.

Our study has several limitations. We simulated a fixed force of infection of SARS-CoV-2 over time among the study population, although this is a simplified approach given SARS-CoV-2 risk is challenging to predict, heterogenous within each age group, changes over time, and variable with emergence of novel variants. Our study was not intended to perform prospective forecasting of COVID-19 outcomes and instead relied on historical estimates. Our study focused primarily on an outcome of severe COVID-19, although we also examined the outcome of clinical cases; we did not evaluate an outcome of long COVID-19 given limited data to inform these estimates. However, it is important to acknowledge that more frequent boosters may reduce risk of cases of long COVID. We relied on surveillance data on COVID-19 outcomes and hospitalizations, which are prone to bias and may overestimate hospitalizations causally attributable to COVID-19. We did not model outcomes in infants, children, or adolescents in the primary analysis, nor did we model risk of myocarditis or any vaccine-related adverse events. We did not simulate booster vaccines more frequent than every 6 months. We include two distinct immunocompromised groups, although acknowledge there is substantial heterogeneity within these populations that is not captured and there is limited data to inform these groups. We did not consider logistical or operational issues with different booster strategies. Finally, we did not account for vaccine hesitancy, which may vary by age group and health conditions[24].

In this study, we find that guidance on frequency of COVID-19 booster vaccination may be strengthened by considering risk of severe COVID-19 defined by age and immunocompromised status to mitigate the burden of COVID-19 in the United States. These results may support guidance decisions on booster timing.

## Methods
### Study population and data
We defined the study population as persons residing in the United States, age 18 years or older, and fully vaccinated (defined as completion of their primary series and 1 or more monovalent booster doses). The epidemiologic data used in the model reflects the timeframe up until approximately September 2022, coinciding with introduction of bivalent COVID-19 vaccines in the United States. Applying publicly available data from the US Centers for Disease Control and Prevention (CDC) COVID-19 surveillance program, we generated age-specific monthly risk estimates of severe COVID-19 (defined as related hospitalization or death)[25,26]. Age-specific seroprevalence estimates were obtained from the CDC based on the nucleocapsid antibody, suggesting prior infection, and updated to account for cases since the last survey[27] (see Appendix, "Prior infection and serosurveillance data").

### Microsimulation model
We developed a stochastic, person-level simulation model (microsimulation) of severe COVID-19 cases in the United States. We created hypothetical cohorts of one million persons in each risk group who were fully vaccinated, defined as having completed their primary series and received at least one monovalent mRNA booster dose. The population size (1 million) for each risk group was chosen to broadly represent the geographic scale of a county in the United States (Table 2). We modeled the population in 12 key risk groups defined by: i) age: 18–49 years, 50–64 years, 65–74 years, 75+ years; and ii) immune status: immunocompetent, mild immunocompromised status (e.g., low-dose corticosteroids, mild immunosuppressive medications), and moderate/severe immunocompromised status (e.g., hematologic malignancy with active treatment or poor response to vaccines, solid organ or bone marrow transplant, high-dose corticosteroids or other moderate/severely immunosuppressive medications)[16] (see Appendix,

**Table 2 | Baseline cohort characteristics and model parameters for severe COVID-19 risk and vaccine effectiveness**

| Cohort characteristics | Model input | Reference |
|---|---|---|
| **Population size (N)** | | |
| Each group | 1 million | |
| **Group** | | |
| Age group | 18–49, 50–64, 65–74, 75+ years | |
| Immune status | Immunocompetent, mild immunocompromised, moderate/severe immunocompromised | |
| **Baseline vaccination status (%)** | | |
| Boosted (1 dose) | 60% | 28 |
| Boosted (2+ doses) | 40% | |
| **Seroprevalence[a] (%)** | | |
| 18–49 years | 82.4% | 27 |
| 50–64 years | 65.8% | |
| 65–74 years | 46.8% | |
| 75+ years | 46.8% | |
| **Severe COVID-19 monthly incidence[b] (cases per 100,000 persons)** | | |
| 18–49 years | 8 | 25 |
| 50–64 years | 16 | 25 |
| 65–74 years | 41 | 25 |
| 75+ years | 113 | 25 |
| **Severe COVID-19 risk ratio for immunocompromised** | | |
| Immunocompromised populations (all) | 2.8 | 25,29,32 |
| **Relative vaccine effectiveness and waning over time (against infection and severe COVID-19)** | | |
| Booster dose | Time-varying (See Appendix) | 1–4, 20, 30 |

[a]Seroprevalence estimated by nucleocapsid antibody to support history of natural infection, with adjustment for number of infections since the last survey to approximate September 2022.
[b]Incidence estimates for severe COVID-19 (defined by hospitalization or death) were generated using publicly available US CDC data, averaging over 6 months preceding September 2022, coinciding with introduction of the bivalent vaccine.
See Appendix for further methodologic description.

"Model calibration"). Upon entry into the simulation, each person was assigned an age, immune status, vaccine status (1 or 2 monovalent mRNA booster doses)[28], and prior infection status[27]. For the age-specific cohorts and the immunocompromised risk group, prior infection status was informed by estimates of seroprevalence (nucleocapsid antibody consistent with prior infection; see Appendix for full methodologic approach)[27,29]. Prior infection status was necessary to define whether an individual had protection from hybrid immunity (vaccine and prior documented infection) or vaccination alone, given that hybrid immunity has been suggested to provide more robust and durable protection compared to vaccination alone[1] (Supplementary Table S1). Each person was assigned a time since their last COVID-19 vaccine or infection (measured in number of months), to account for waning of protection over time. This timing was determined from sampling of publicly available data on time series data of vaccine administration and COVID-19 cases and then tracked over the simulation period (Supplementary Fig. S4).

We simulated a two-year time horizon, which was chosen to allow adequate time for comparison of vaccine strategies (i.e., one year time horizon would not allow estimation of differences from one-time and annual strategies). We assumed a hypothetical fixed population with no aging or demography. The start of the simulation (time 0) coincided with approximately September 2022, alongside introduction of the bivalent vaccine in the United States.

During the simulation, we applied an individual-specific, time-varying probability of SARS-CoV-2 infection and severe COVID-19 for

each month time step, informed by the model calibration using COVID-19 surveillance datasets (see Calibration and Validation section). This probability combined a fixed group-specific 'force of infection' term by age and immune status and an individual, time-varying level of protection against SARS-CoV-2 infection and severe COVID-19. An individual's risk of SARS-CoV-2 infection and severe disease changed over time as protection waned. The primary analysis used a static model of infection, meaning we did not account for indirect effects due to vaccination (i.e., reduced transmission due to vaccine-induced protection), although we did test a dynamic transmission model in an alternative analysis (see Scenario Analysis). Each person's level of protection was based on vaccine status (time since last vaccine) and prior infection history (time since last infection, if applicable). This model explicitly accounted for waning of protection against SARS-CoV-2 infection and severe COVID-19 independently based on timing of last vaccination and prior infection, which was estimated from literature[1–3,5,20,30] (Supplementary Tables S1-S2). We separately modeled individuals as either having vaccine-induced (without prior infection) or hybrid immunity (defined as vaccination with documented prior infection) since literature suggests far higher and more durable protection for hybrid immunity[1,20] (Supplementary Fig. S1).

We simulated severe COVID-19 cases, defined as a composite outcome of COVID-19 related hospitalization or COVID-19 related death. The study focused primarily on severe COVID-19 based on a public health priority to reduce hospitalizations and deaths, although we did simulate non-severe COVID-19 cases and subsequent effects on protection and immunity (Supplementary Table S9). All COVID-19 cases (severe and non-severe) reset the time since last COVID-19 case or vaccine. While acknowledging that a certain fraction of COVID-19 cases will result in long COVID, we did not account for long COVID given limited data to inform these estimates. We assumed no reinfections occurred within 90 days of a SARS-CoV-2 infection. Analysis was conducted in R (version 4.2.1).

### Vaccination strategies

We simulated three distinct vaccination strategies with booster vaccines for COVID-19, including: i) one-time booster at the start of the simulation (base case); ii) single booster followed by annual boosters (total of 2 doses); and iii) single booster followed by boosters every 6 months (semiannual; total of 4 doses). In September 2022, the available COVID-19 booster vaccine in the United States was the bivalent vaccine (ancestral strain and Omicron subvariants BA.4/5), followed later by a monovalent formulation against Omicron XBB.1.5. Each round of vaccination was administered in the population over a 3-month period. We calibrated the protection and waning of a mRNA booster dose to published data on vaccine effectiveness over time using data from both monovalent and bivalent COVID-19 booster vaccine literature (Supplementary Table S1)[1–3,5]. We modeled the benefit of a booster dose to restore maximal protection against severe COVID-19 prior to waning (Supplementary Fig. S1). Therefore, the impact of additional vaccination conservatively did not increase the absolute protective effectiveness previously achieved, but only restored the lost protection due to waning. This approach to vaccine modeling resulted in estimates of relative vaccine effectiveness similar to published estimates on the bivalent mRNA booster (Supplementary Fig. S3)[3]. We estimated the waning protective effectiveness of a booster dose by age group and prior infection status over a 24-month period using a linear mixed effects model. We modeled the outcome of protection against severe COVID-19 and infection as the log of 1 minus protective effectiveness, with predictor variables of the log of months since last vaccine dose or COVID-19 illness (whichever was more recent), age group (18–49 years, 50–64 years, 65+ years), and prior infection status, based on available literature. We modeled two immunocompromised groups, generating age-specific estimates for a

mild immunocompromised group (13% lower protection) and moderate or severe immunocompromised group (25% lower protection, incorporating faster waning)[2,16,31,32]. We assumed that each repeated booster dose would achieve the same level of effectiveness without immune exhaustion, immune imprinting phenomenon, or reduced vaccine effectiveness due to new variants[33,34], although we explored this in sensitivity analyses.

### Study Outcomes

The primary study outcome was severe COVID-19, measured as the annual absolute risk of severe COVID-19 over a 2-year simulation period in each risk group. Each of the boosting strategies was compared to the base case of a one-time booster at the start of the simulation. For each strategy, we estimated the total number of severe COVID-19 cases, absolute annual risk reduction of severe COVID-19 (cases per 100,000 persons), relative risk reduction, and NNT with a specified vaccination frequency to avert one severe COVID-19 case (calculated per person, not vaccine dose).

### Calibration and validation

We calibrated the model to age-specific estimates of severe COVID-19 risk generated from an average over the 6-month period preceding model initialization (March 2022–August 2022). For the two immunocompromised populations, we used literature estimates for their age distribution, assuming the same age-specific risk of infection but 2.8-fold higher risk of severe disease given infection[25,29,32] (see Appendix "Model calibration"; see Table 2 for severe COVID-19 risk estimates). This calibration yielded a per month, 'force of infection' coefficient specific to each age and immune status on their risk of severe COVID-19, which was multiplied against 1 minus an individual's current level of protection to obtain individual per month probability of severe COVID-19. The probability of SARS-CoV-2 infection (non-severe) was modeled with an additional multiplier and separate estimates on level of protection (see Appendix, "Model calibration"). For model validation, we performed a comparison of model-predicted outcomes over the first 3 months of the simulation (September 2022-November 2022).

### Scenario analysis: simulation of novel variants

We repeated the primary analysis under different scenarios for emergence of novel variants with immune evasion (Fig. 1A), including one scenario with a variant targeted vaccine. Upon circulation of a novel variant, we modeled two different immune evasion scenarios: i) absolute protection from vaccine or hybrid protection against non-severe and severe COVID-19 is reduced by 10%, due to immune evasion; and ii) absolute protection is reduced by 10%, and rate of waning increases by 5%. We did not simulate variants with higher infectiousness or severity. In the scenario with a variant targeted vaccine, we assumed the vaccine restored the protection lost due to the new variant in vaccine-induced immunity and partially restored protection for hybrid immunity. Novel variants were introduced over a 3-month period. A full description of the analysis is available in the Appendix (see "Scenario analysis: Novel variants").

### Scenario analysis: dynamic transmission model

We repeated the primary analysis using a dynamic transmission model, which accounted for the indirect effects of vaccination on transmission. This analysis was designed to test the importance of considering transmission dynamics in the analysis. This model departed from the primary microsimulation model based on the following modifications. First, the 'force of infection' term was formulated to be directly related to the number of SARS-CoV-2 infections in the population in the prior time step (week) with age-specific contact matrices[35,36]. Second, the simulated population included all age groups and unvaccinated individuals. Third, vaccine strategies were applied with imperfect uptake

coverage by age- and immune status to reflect current uptake (Supplementary Table S7). Fourth, the model was only calibrated to match observed severe COVID-19 cases at time 0 (Supplementary Table S8). We compared booster vaccination strategies in the following groups to determine the impact of indirect effects of vaccination: i) 75+ years and moderate/severe immunocompromised; ii) 65+ years and mild and moderate/severe immunocompromised; and iii) all groups 18+ years. In all strategies, we applied one-time booster vaccination as the base case intervention to those 18+ years based on expected uptake. Study outcomes were computed among persons assigned to the booster vaccination strategies (i.e., excluding unvaccinated persons, or those who did not receive additional vaccination), to improve comparability to the primary model. A full description of the model specifications is available in the Appendix (see "Scenario analysis: Dynamic transmission model").

### Sensitivity analysis and uncertainty

We conducted sensitivity analyses on the main microsimulation analysis to evaluate the robustness of our findings. First, we repeated the analysis under optimistic or pessimistic assumptions on level of protection (10% lower or higher) from vaccine-induced and hybrid immunity, as well as differential waning of protection (10% lower or higher) (Supplementary Tables S13–S16). Second, we repeated the analysis for a lower (0.5x) and higher (2x) incidence of severe COVID-19 (Supplementary Tables S17, S18). Third, we performed analyses under the assumption that additional boosters would have lower vaccine effectiveness (i.e., immune exhaustion) (Supplementary Table S25). Fourth, we performed the analysis with higher or lower seroprevalence and an additional analysis with a population of only previously infected persons (i.e., 100% seroprevalence) (Supplementary Tables S19–S21). Fifth, we repeated the analysis assuming higher proportion of sub-clinical infections (Supplementary Table S24). Additional details on sensitivity analyses can be found in the Appendix (see "Sensitivity analysis").

We generated uncertainty intervals for the primary analysis based on parameter uncertainty in vaccine effectiveness and waning over time, baseline seroprevalence levels, and non-severe infection multipliers (Supplementary Table S5). This interval is generated by simulating the full range of model inputs at baseline, which define the bounds of the interval; the reported point estimate uses the base case assumption of model inputs, so the bounds are expected to be asymmetric relative to the point estimate. Uncertainty intervals are designed to demonstrate uncertainty within a single vaccine strategy under a range of baseline conditions; vaccine strategies should be compared against one another using the same set of assumed baseline model inputs.

### Ethical approval

This study was not human subjects research given use of publicly available secondary datasets with aggregated estimates that are not identifiable.

### Reporting summary

Further information on research design is available in the Nature Portfolio Reporting Summary linked to this article.

## Data availability

This study used publicly available secondary datasets from the US Centers for Disease Control and Prevention[25–27]. More details on these datasets can be found in the Appendix.

## Code availability

Analytic code is available at: https://github.com/hailey-park/booster-timing[37].

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

## Acknowledgements

This study is supported by a National Institutes of Health, NIAID New Innovator Award (DP2 AI170485). This study was supported by funding from the California Department of Public Health. The funding source had no role in the design, conduct, or analysis of this study or in the decision to submit the manuscript for publication. The findings and conclusions in this article are those of the authors and do not necessarily represent the views or opinions of the California Department of Public Health, California Health and Human Services Agency, or National Institutes of Health.

## Author contributions

Ms. Hailey Park and Dr. Nathan Lo had full access to all the data in the study and take responsibility for the integrity of the data and the accuracy of the data analysis. Study concept and design: RMW, RS, NCL; Statistical analysis: HJP, STT, NCL; Analytic coding: HJP, STT; Input on statistical analysis: GSG, ADP; Acquisition, analysis, or interpretation of data: HJP, GSG, STT, JDK, GWR, RMW, RS, ADP, NCL; First draft of the manuscript: HJP, NCL; Critical revision of the manuscript: HJP, GSG, STT, JDK, GWR, RMW, RS, ADP, NCL; Contributed intellectual material and approved final draft: HJP, GSG, STT, JDK, RS, GWR, RMW, ADP, NCL.

## Competing interests

NCL reports consulting fees from the World Health Organization related to guidelines on neglected tropical diseases, which are outside the scope of the present work. The remaining authors have no conflicts to declare.
