## [Peer Review File · Nature Communications]

Comparing frequency of booster vaccination to prevent severe COVID-19 by risk group in the United StatesREVIEWER COMMENTS

Reviewer #1 (Remarks to the Author):

The paper introduces an interesting comparative analysis, utilizing modeling estimates, to examine the advantages of various COVID-19 bivalent booster dose allocations. The research question is both relevant and intriguing. However, I have a few observations regarding the study that I would like to be addressed. My primary concern revolves around the appropriateness of the proposed methodology in effectively addressing the research question.

Major comments

1) The proposed model is able to simulate the risk of severe COVID-19 based on various individual features, such as vaccination status, age, and prior infection. However, a major limitation of the study is that it does not take into account transmission and the underlying spreading dynamics. This limitation may challenge the validity of some results. According to my understanding, the model assumes a fixed risk of infection over a two-year time period. What is the assumption behind this choice? From my point of view such a situation is never met in practice. More concerning, the probability of COVID-19 infection is influenced by the vaccination rollout. As individuals receive booster doses, the probability of infection will be affected and will decrease (individuals are indeed increasingly less susceptible and infectious). Additionally, this aspect may introduce bias into the presented results. For instance, by vaccinating younger population groups who are generally more active and contribute more to the spread, an indirect effect on severity may be achieved. In other words, an overall reduction in severity among the more at-risk population could be attained by reducing infections through vaccinating the population that spreads the virus more. Based on my understanding, the current model overlooks these crucial aspects. Consequently, the results mainly stem from the assumed conditions, rather than from simulated dynamics. Essentially, the paper concludes that individuals at lower risk of severe COVID-19 are indeed less susceptible to severe complications from the disease, and that more frequent booster doses that restore protection are beneficial. To enhance the paper's quality and significance, I believe it is crucial to incorporate transmission dynamics into the model or alternatively to show that the assumptions made are valid and do not challenge the significance of results.

2) I find the title and introduction of the paper to be a bit misleading. The title mentions a "prediction" aspect that is not adequately explained or addressed within the text. Additionally, the introduction presents the problem as determining the optimal timing of booster doses. However, this objective is not effectively achieved in the paper, as it primarily focuses on comparing different scenarios and analyzing a limited set of output quantities. Consequently, the paper does not sufficiently address the optimal timing problem it initially presents. To improve clarity and alignment between the title, introduction, and the actual content of the paper, I think it is important to revise both the title and the introduction.

3) The introduction lacks a discussion on previous literature on the subject. There have been numerous modeling studies focusing on understanding and comparing various vaccine allocation strategies for the primary vaccination cycle. It would be beneficial for the authors to reference and discuss some of these approaches, as their paper can be seen as a natural extension of these efforts, but applied to the later phases of the COVID vaccination campaign. To provide some examples, here are a few relevant papers that the authors may find interesting to explore and potentially include in their literature review:

<https://www.science.org/doi/10.1126/sciadv.abf1374>

<https://www.science.org/doi/10.1126/science.abe6959>

<https://journals.plos.org/ploscompbiol/article?id=10.1371/journal.pcbi.1009346>

<https://www.nature.com/articles/s41591-021-01334-5>

Minor comments

4) To ensure clarity and comprehension for readers from diverse disciplines, I would advise to define abbreviations upon their first usage in the text. In the current manuscript, certain abbreviations are introduced only in the Material and Methods section, making it challenging for

readers to grasp their meaning. (for example, I could not understand the meaning of "NNT" until I reached the material and methods section)

5) Similarly, it is never mentioned what a "bivalent" booster is, so it would be good to briefly mention it in the introduction.

6) I think some of the methods presented in the Supplementary Information need to be brought to the methods section of the main manuscript, as they are key to understanding the paper. First of all, how the waning is modelled. This is a fundamental assumption underlying all findings, so I think it should be discussed in the main text.

7) The readability of the plots can be improved by making adjustments to their design. The current plots consisting of white panels and small points make it challenging for readers to interpret the data effectively. A few suggestions would be to increase the size of data points, and adding y-axis gridlines.

8) An additional limitation of the model is the absence of consideration for vaccine hesitancy, which can vary among different age groups and health conditions. This omission can have implications for the effectiveness of specific vaccine scenarios, so I believe it is worth at least mentioning this aspect in the Discussion.

9) It would be beneficial to have in the discussion or in the introduction a more in-depth discussion of the methodology used, mentioning if microsimulations have been used in other contexts of COVID-19, what are the pros and cons and relevant references.

10) The abstract reads "annual and semiannual bivalent boosters would reduce annual absolute risk of severe COVID-19 by 311 (277-369) and 578 (494-671) cases, respectively, compared to a one-time bivalent booster dose.", I think these numbers are per 100,000 individuals, it would be more clear to mention this explicitly.

11) Page 14: "We estimated this risk for persons vaccinated with the primary series and at least one monovalent booster This calibration yielded..." I think a "." is missing between "booster" and "This".

Reviewer #2 (Remarks to the Author):

The authors present a very clear and well-structured analysis of the respective benefits of different vaccination frequencies in preventing severe COVID-19 disease amongst different cohorts in the US population. The results are based on current vaccines in use in the US against Omicron type infection characteristics collated from a variety of existing sources and applied using a simple model for immunity from disease at an individual level.

There are however a number of limitations to the approach that affect the use of the results in providing a reliable future forecast. Firstly, vaccine effects are only considered at a personal and not population level. Since the study only considers efficacy versus severe disease effects, and not in limiting susceptibility/transmissibility/symptom prevalence, the use of vaccines in limiting infection spread is not considered— something which was considered to be very important when vaccines for COVID were first introduced. This is perhaps okay in a situation where future vaccination is concentrated only on protecting the vulnerable, but certainly needs more discussion.

Secondly, due to continued high levels of infection, COVID-19 is still rapidly evolving and immunity from both previous infection and current vaccines is highly volatile. It is likely that vaccines will continued to be adapted to meet these challenges (in a similar manner to seasonal influenza), but COVID has yet to settle into endemicity, leaving high levels of uncertainty. While the authors do a good job in presenting parameter sensitivity, future disease could easily be expected to fall outside of their predictions.

Finally, the study is inherently US specific. Vaccine use (type and prevalence), healthcare, and demographics vary greatly across the world, all of which greatly affect the occurrence of severe disease. Data sources and fitting are exclusively US specific, and I suggest the authors should consider adjusting the title to reflect this.

Despite these limitations, the article is commendable in presenting some valuable insights into long-term COVID-19 vaccine effects, which is worthy of publication. However, I believe the limitations mentioned above need more careful consideration if they are to feature in a high impact journal.

Reviewer #3 (Remarks to the Author):

SUMMARY

This manuscript estimates the health benefit (in terms of averted cases of severe COVID-19) of different frequencies of booster vaccination for different population groups. I have some questions about the set-up of the scenarios that are investigated in the paper, and some smaller comments.

MAJOR ISSUES

- For the analysis of immunocompromised individuals, I worry that looking at a single group simplified the analysis too much. As the extent of immune deficiency will differ between health conditions, I would expect the benefits of vaccination would also differ condition by condition. Can this be reflected in the paper (preferably in analysis or at least in discussion)?
- The results are presented as providing information on the individual benefits of vaccination, but do not appear to stratify results by information that an individual presumably would have – that is, the recency of prior vaccination and infection. As this information would impact an individual's existing level of immunity, shouldn't this be considered with individuals make vaccination decisions?
- Why was the immunocompromised group not stratified by age? Naively, one might conclude a 75-year-old with HIV infection (220 per 100,000) is at lower risk than a 75-year-old without HIV infection (311 per 100,000). Clearly this is not the intended interpretation, but the current results format makes it hard to understand risk for immunocompromised individuals at different ages, yet it is known that age is a big risk factor.
- I am surprised that the sensitivity analyses presented in the main text do not include the introduction of a new variant with substantial immune escape – would this have an effect on the results? On page 13 it is mentioned that this is explored in sensitivity analyses, but these results are not seen in Figure 1.

MINOR ISSUES

- P2: I suggest adding “per 100,000 persons” for the first set of quantitative results reported (“reduce annual absolute risk of severe COVID-19 by 311 (277-369) and 578 (494-671) cases”), otherwise it seems like these are results for the whole US population, not per 100,000.
- P4, paragraph 3: The following is confusing to me: “the model estimated 2,158 severe COVID-19 cases over a 2-year period in a population of 1 million people with annual risk of 108 severe cases per 100,000 persons (95% UI: 105-108).” It seems like this is basically the same statistic, just reported in different units?
- P5, paragraph 1: “without prior documented COVID-19 infection” would clarify whether this was from the start of the simulation, or at the point of developing severe disease (for example, consider someone infected in year 1, and then again in year 2, and developed severe disease in year 2).
- P6, paragraph 3: “Additional results for sensitivity analysis are available in the Appendix.” I

would suggest summarizing these results in the main text, even if this needs to be done succinctly.

- P7, paragraph 1: For the statement “We found ... than a single recommendation for the general population” I suggest rewording this – as this was not a comparison that was done in the analysis I am not sure this should be presented as a conclusion of the study as it currently is.
- P8, paragraph 1: A picky point: for the text “influenza vaccination to prevent death (NNT 48)” I worry about providing this number for comparison, as it seems an extremely low estimate of the NNT for this intervention and outcome (e.g. would need vaccination to reduce flu risk from 100% to 0%, and for flu to have an IFR of 2%, or some similar combination of assumptions). I realize that this NNT comes from a cited study, but still it stands out.
- Figure A1 is so central to the analysis it would be great if there were room for it in the main text.
- Table A2: the table note reads “stochastic variation explains the discrepancy between model and observed output”. Can you include intervals for the model output to confirm this? For 75+ and Immunocompromised groups the difference in incidence is a bit concerning.
- Figure A4: apologies if I am understanding the quantities being plotted – is the line the central estimate (eg, mean, or median), and are the shaded areas the interval? If so, why does the central estimate fall outside the interval in some situations? I have the same concern about Figure S1. Also, some of the intervals in the tables appear to suffer the same problem (from Table S4, page 18, we see intervals 1,337 (1,321 – 1,337) and 193 (176 – 194), which don’t seem right.

Park *et al.* “Comparison of timing of booster vaccination for COVID-19 to prevent severe disease by risk group in the United States”

(Reference no. NCOMMS-23-29382) – Point-by-point response

Response to Reviewer 1:

General comment

The paper introduces an interesting comparative analysis, utilizing modeling estimates, to examine the advantages of various COVID-19 bivalent booster dose allocations. The research question is both relevant and intriguing. However, I have a few observations regarding the study that I would like to be addressed. My primary concern revolves around the appropriateness of the proposed methodology in effectively addressing the research question.

Response: We appreciate the constructive and supportive comments. To address this primary concern, we have added a new scenario analysis using a dynamic transmission model as suggested, which is detailed below.

Comment 1

MAJOR COMMENT

1) The proposed model is able to simulate the risk of severe COVID-19 based on various individual features, such as vaccination status, age, and prior infection. However, a major limitation of the study is that it does not take into account transmission and the underlying spreading dynamics. This limitation may challenge the validity of some results. According to my understanding, the model assumes a fixed risk of infection over a two-year time period. What is the assumption behind this choice? From my point of view such a situation is never met in practice. More concerning, the probability of COVID-19 infection is influenced by the vaccination rollout. As individuals receive booster doses, the probability of infection will be affected and will decrease (individuals are indeed increasingly less susceptible and infectious). Additionally, this aspect may introduce bias into the presented results. For instance, by vaccinating younger population groups who are generally more active and contribute more to the spread, an indirect effect on severity may be achieved. In other words, an overall reduction in severity among the more at-risk population could be attained by reducing infections through vaccinating the population that spreads the virus more. Based on my understanding, the current model overlooks these crucial aspects. Consequently, the results mainly stem from the assumed conditions, rather than from simulated dynamics. Essentially, the paper concludes that individuals at lower risk of severe COVID-19 are indeed less susceptible to severe complications from the disease, and that more frequent booster doses that restore protection are beneficial. To enhance the paper's quality and significance, I believe it is crucial to incorporate transmission dynamics into the model or alternatively to show that the assumptions made are valid and do not challenge the significance of results.

Response: We appreciate the reviewer's comment on the importance of considering transmission dynamics in the analysis by accounting for the indirect effects of vaccination. To clarify, in the original model, an individual's risk of both SARS-CoV-2 infection and severe disease changes over time as protection wanes and additional vaccination reduces both the risk of infection and severe disease, but the overall “force of infection” was fixed (therefore did not account for indirect effects of vaccination on transmission). We clarify this in the Methods section (see changes below). However, we agree with the reviewer about considering indirect effects of vaccination and have therefore **added a new analysis using a dynamic transmission model**, which accounts for the

indirect effects of vaccination on transmission dynamics. We include this as a separate and complementary analysis to the original model (see new Figure 3). The Methodology is described below. Specifically, we compare frequent booster vaccine strategies focused only on the highest risk persons (75+ years, moderate/severe immunocompromised group) versus a more inclusive strategy (18+ years in all groups), expecting that indirect effects would be more pronounced with a more inclusive strategy, as queried by the reviewer. This analysis is designed to test the importance of considering transmission dynamics in the analysis.

While the dynamic model demonstrates some modest differences in impact from indirect effects attributable to more frequent booster vaccination with a more inclusive strategy (18+ in all age groups) compared to a targeted strategy (75+ years, severely immunocompromised only), within the assumed conditions, the overall model conclusions are broadly similar. This is likely since: i) booster vaccines have indirect protection, but they are relatively short-lived and modest; ii) vaccine booster uptake is likely to be low within the population based on current observed coverage; iii) a substantial proportion of the population remains unvaccinated or under vaccinated. Therefore, under realistic assumptions, the indirect effects of booster vaccination are likely to be modest given low uptake of vaccines and short-lived indirect effects of booster vaccines. These conditions all favor that transmission dynamics are unlikely to change our study's overall conclusions. We have included the new analysis in the study and additional new text to the Discussion section to describe these findings.

We have kept the static model as the main analysis for several reasons. First, as we demonstrate with the new sensitivity analysis using a dynamic transmission model, the static model is a valid modeling approach to answer the study question. Second, the static model allows us to better calibrate to observed data on severe COVID-19 cases, seroprevalence, and other key parameters, which are challenging to do with a dynamic model. Third, the dynamic transmission model forces our model to make many additional assumptions that have limited data, such as age-based mixing and transmission amongst others. Fourth, the public health and clinical question of our study focuses on vaccinated populations and adults; however, the dynamic transmission model requires modeling children and unvaccinated persons and anticipated coverage, which are uncertain. Therefore, we have kept the original model as the main analysis but do include the new dynamic transmission model suggested by the reviewer. The methodological details and results of the new analysis using a dynamic transmission model are outlined here. We also add additional Discussion about this important consideration. Our new results support a similar conclusion.

In Methods (Clarification to main model):

“During the simulation, we applied an **individual-specific, time-varying** probability of SARS-CoV-2 infection and severe COVID-19 for each month time step, informed by the model calibration using COVID-19 surveillance datasets (see Calibration and Validation section). This probability combined a **fixed group-specific ‘force of infection’ term by age and immune status** and an individual, **time-varying level of protection against SARS-CoV-2 infection and severe COVID-19. An individual’s risk of SARS-CoV-2 infection and severe disease changed over time as protection waned. The primary analysis used a static model of infection, meaning we did not account for indirect effects due to vaccination (i.e., reduced transmission due to vaccine-induced protection), although we did test a dynamic transmission model in an alternative analysis (see Scenario Analysis).”**

In Methods (Scenario analysis: Dynamic transmission model):

“Scenario analysis: Dynamic transmission model

We repeated the primary analysis using a dynamic transmission model, which accounted for the indirect effects of vaccination on transmission. This analysis is designed to test the importance of considering transmission dynamics in the analysis. This model departed from the primary microsimulation model based on the following modifications. First, the ‘force of infection’ term was formulated to be directly related to the number of SARS-CoV-2 infections in the population in the prior time step (week). Second, the simulated population included all age groups and unvaccinated individuals. Third, vaccine strategies were applied with imperfect uptake coverage by age- and immune status to reflect current uptake (see Appendix, Table A6). Fourth, the model was only calibrated to match observed severe COVID-19 cases at time 0 (see Appendix). We compared booster vaccination strategies in the following groups to determine the impact of indirect effects of vaccination: i) 75+ years and moderate/severe immunocompromised; ii) 65+ years and mild and moderate/severe immunocompromised; and iii) all groups 18+ years. In all strategies, we applied one-time booster vaccination as the base case intervention to those 18+ years based on expected uptake (see Appendix). Study outcomes were computed among persons assigned to the booster vaccination strategies (i.e., excluding unvaccinated persons, or those who did not receive additional vaccination), to improve comparability to the primary model. A full description of the model specifications is available in the Appendix.”

In Appendix (Technical Appendix):

“Scenario analysis: Dynamic transmission model

In this scenario analysis, we repeated the primary analysis using a dynamic transmission model, which accounted for the indirect effects of vaccination on transmission. The objective was to determine to what extent different booster vaccination strategies affected transmission and, by extension, risk of severe COVID-19. The dynamic model had key modifications from the primary microsimulation model. First, the ‘force of infection’ term was formulated to be directly related to the number of SARS-CoV-2 infections in the population in the prior time step (week). This additional term of $\frac{I_{t-1}}{N}$ was applied to estimate the probability of SARS-CoV-2 infection and severe COVID-19, where I_{t-1} is the number of infections during the prior week and N is the population size. The model was changed to be on a week time step and re-calibrated. Second, the simulated population included all age groups (addition of children, 0-17 years) and unvaccinated individuals. We used 100% minus the age-specific coverage estimates of primary series completion to estimate the proportion of unvaccinated persons¹¹. Third, vaccine strategies were applied with imperfect uptake coverage by age- and immune status to reflect current values (see Appendix, Table A6). Table A6 includes estimates for age-specific coverage of vaccine uptake. Fourth, we simulated a total population of 10 million, ensuring that age- and immunocompromised status reflected the United States population. Table A6 includes the assumed demography and risk of being immunocompromised. We assumed equal mixing between age groups and homogenous mixing overall. Fifth, while the model was calibrated to match observed severe COVID-19 outcomes at baseline (time 0), the model was not calibrated to match a defined number of severe COVID-19 cases over the 2-year simulation period. Under the described approach to calibration, the dynamic model estimated a higher number of severe COVID-19 cases in the base case model compared to the static model over the entire simulation period. Since the goal of this model was not to predict the trends in COVID-19 outcomes over time, but rather compare the potential impact of indirect effects

under different vaccine strategies by risk group, this approach to calibration was kept to minimize introduction of additional assumptions. Overall, the relative comparison between risk groups under different vaccine strategies was more important than the absolute estimates of severe COVID-19 risk to determine the potential impact of indirect effects. We compared booster vaccination strategies in the following groups to determine the impact of indirect effects of vaccination: i) 75+ years and moderate/severe immunocompromised; ii) 65+ years and mild and moderate/severe immunocompromised; and iii) all groups 18+ years. We compared these booster vaccination strategies under two uptake scenarios: i) realistic uptake modeling current up-to-date coverage of boosters; and ii) optimistic uptake with higher coverage. In all strategies, we applied one-time booster vaccination with expected coverage from Table A6. The largest indirect effects from vaccination are expected with more inclusive vaccine strategies and optimistic coverage. Study outcomes were computed in among persons assigned to the booster vaccination strategies (i.e., excluding unvaccinated persons, or those who did not receive additional vaccination); this was done to improve comparability to the primary model.”

In Results:

“To investigate the impact of indirect effects of vaccination on transmission, we repeated the primary analysis using a dynamic transmission model (Figure 3). We found that indirect effects were larger with more inclusive frequent booster vaccine strategies, although within the assumed conditions, the overall model conclusions were broadly similar to the static model. In persons 75+ years old, the dynamic model estimated that semiannual booster vaccination would lead to an annual risk reduction of 302 severe cases per 100,000 persons with focused vaccination (75+ years and moderate/severe immunocompromised groups) and reduction of 406 severe cases per 100,000 persons with more inclusive vaccination (18+ years all groups) compared to a one-time booster vaccination.”

In Discussion:

“The primary analysis did not account for the indirect effects of more frequent booster vaccination on transmission.²⁰⁻²² To evaluate the importance of indirect effects on our study findings, we performed a scenario analysis using a dynamic transmission model. While we found some differences due to indirect effects from more frequent booster vaccination in more inclusive strategies (e.g., 18+ years in all groups), the overall model conclusions were similar. This is likely since: i) booster vaccines yield indirect protection, but these effects are relatively short-lived and modest²⁰⁻²²; ii) vaccine booster uptake is overall low within the population based on current coverage estimates; iii) a substantial proportion of the population remains unvaccinated or under vaccinated. Therefore, under reasonable assumptions, the indirect effects of booster vaccination are unlikely to change the study conclusions that consider direct protection alone. This reinforces the validity of using a static model, which relies on fewer assumptions, for the primary analysis.”

In Tables and Figures:

Figure 3: Scenario analysis using a dynamic transmission model to estimate the impact of indirect effects on COVID-19 booster vaccination strategies in the 75 years and older group.

We used a dynamic transmission model to compare different frequencies of COVID-19 booster vaccine in the following groups: (A) 18+ years in all groups (most inclusive); (B) 65+ years and all immunocompromised groups; and (C) 75+ years, moderate/severe immunocompromised group (most restrictive). We assumed a background of one-time booster vaccination at the start of the simulation in adults (18+ years) with age-specific, imperfect vaccine uptake. We plotted absolute annual risk of severe COVID-19 over a two-year simulation in the 75+ year risk group, to compare the indirect effects of booster vaccination on this high risk group. The vertical bars represent uncertainty intervals, which simulate different scenarios of baseline conditions to account for uncertain model inputs. Intervals are designed to demonstrate uncertainty within a single vaccine strategy rather than for comparison between vaccine strategies; vaccine strategies should be compared using the same assumed baseline conditions. Estimates for additional risk groups and alternative uptake rates are available in the Appendix.

Comment 2

2) I find the title and introduction of the paper to be a bit misleading. The title mentions a "prediction" aspect that is not adequately explained or addressed within the text. Additionally, the introduction presents the problem as determining the optimal timing of booster doses. However, this objective is not effectively achieved in the paper, as it primarily focuses on comparing different scenarios and analyzing a limited set of output quantities. Consequently, the paper does not sufficiently address the optimal timing problem it initially presents. To improve clarity and alignment between the title, introduction, and the actual content of the paper, I think it is important to revise both the title and the introduction.

Response: We have revised the Title and Introduction as suggested (shown below). We agree with the reviewer that our study compares a limited set of options for frequency of booster vaccination. To address this point, we have revised the title to remove prediction (see revision below). We have also edited the Introduction and entire manuscript to remove “optimal” and instead state a goal of measuring the comparative effectiveness of different frequencies of booster vaccination.

In Title:

“Comparison of timing of booster vaccination for COVID-19 to prevent severe disease by risk group in the United States”

In Introduction:

“A key question remains: **what is the optimal comparative effectiveness of different frequencies** of COVID-19 booster vaccination in key risk groups to offset waning of protection against severe disease?”

“Given heterogeneity in risk of severe COVID-19 within the population, **the optimal comparative effectiveness of different frequencies of COVID-19 booster vaccination** may vary based on key risk factors.”

In Abstract:

“This highlights a broader need to understand **how optimal different timing** of COVID-19 booster vaccines may mitigate the risk of severe COVID-19, while accounting for waning of protection and differential risk by age and immune status.”

Comment 3

3) The introduction lacks a discussion on previous literature on the subject. There have been numerous modeling studies focusing on understanding and comparing various vaccine allocation strategies for the primary vaccination cycle. It would be beneficial for the authors to reference and discuss some of these approaches, as their paper can be seen as a natural extension of these efforts, but applied to the later phases of the COVID vaccination campaign. To provide some examples, here are a few relevant papers that the authors may find interesting to explore and potentially include in their literature review:

<https://www.science.org/doi/10.1126/sciadv.abf1374>

<https://www.science.org/doi/10.1126/science.abe6959>

<https://journals.plos.org/ploscompbiol/article?id=10.1371/journal.pcbi.1009346>

<https://www.nature.com/articles/s41591-021-01334-5>

Response: We appreciate the suggestion to include a broader literature to improve the framing of the study and consider their conclusions in our work. We have revised the introduction to include the suggested references and describe these studies as prior work to guide our modeling approach.

In Introduction:

“**While there was considerable study of vaccine prioritization during introduction of the COVID-19 vaccine⁹⁻¹³**, there is limited evidence to guide decisions on the timing of COVID-19 booster vaccination to prevent severe COVID-19. **Considerations** to determine the frequency of COVID-19 booster vaccination for an individual include...”

In Discussion:

“Most of the estimated benefit from more frequent booster vaccination occurred in older age groups, the immunocompromised, and those without prior COVID-19, **which is consistent with prior literature analyzing vaccine prioritization during introduction of COVID-19 vaccines^{9,10,13}**.”

In References:

9. Bubar, K.M. et al. Model-informed COVID-19 vaccine prioritization strategies by age and serostatus. *Science* 371, 916-921 (2021).

10. Chapman, L.A.C. et al. Risk factor targeting for vaccine prioritization during the COVID-19 pandemic. *Sci Rep* 12, 3055 (2022).

11. Giordano, G. et al. Modeling vaccination rollouts, SARS-CoV-2 variants and the requirement for non-pharmaceutical interventions in Italy. *Nat Med* 27, 993-998 (2021).
12. Gozzi, N., Bajardi, P. & Perra, N. The importance of non-pharmaceutical interventions during the COVID-19 vaccine rollout. *PLoS Comput Biol* 17, e1009346 (2021).
13. Matrajt, L., Eaton, J., Leung, T. & Brown, E.R. Vaccine optimization for COVID-19: Who to vaccinate first? *Sci Adv* 7(2020).

Comment 4

MINOR COMMENTS

1) To ensure clarity and comprehension for readers from diverse disciplines, I would advise to define abbreviations upon their first usage in the text. In the current manuscript, certain abbreviations are introduced only in the Material and Methods section, making it challenging for readers to grasp their meaning. (for example, I could not understand the meaning of “NNT” until I reached the material and methods section)

Response: We agree and have defined NNT during its first appearance in the Results section.

Comment 5

2) Similarly, it is never mentioned what a “bivalent” booster is, so it would be good to briefly mention it in the introduction.

Response: We have clarified this in the Introduction.

In Introduction:

“Considerations to determine the frequency of COVID-19 booster vaccination (**e.g., bivalent vaccines targeting more than one variant, such as the ancestral strain and Omicron subvariants BA.4/5; or monovalent vaccines targeting one variant such as Omicron XBB.1.5**) for an individual include:...”

Comment 6

3) I think some of the methods presented in the Supplementary Information need to be brought to the methods section of the main manuscript, as they are key to understanding the paper. First of all, how the waning is modelled. This is a fundamental assumption underlying all findings, so I think it should be discussed in the main text.

Response: We appreciate the reviewer’s suggestion. We have moved the section describing our approach to modeling estimation of protective effectiveness and its waning from the Appendix into the Methods section in the main text.

In Methods (Vaccine Strategies):

“We calibrated the protection and waning of a mRNA booster dose to published data on vaccine effectiveness over time using data from both monovalent and bivalent COVID-19 booster vaccine literature during Omicron variant predominance (Table A1).^{1-3,5} **We modeled the benefit of a booster dose to restore maximal protection against severe COVID-19 prior to waning (see Appendix, Figure A1). Therefore, the impact of additional vaccination conservatively did not increase the absolute protective effectiveness previously achieved, but only restored the lost protection due to waning. This approach to vaccine modeling resulted in estimates of relative vaccine effectiveness similar to published estimates on the bivalent mRNA booster**

(see Figure A2)³. We estimated the waning protective effectiveness of a booster dose by age group and prior infection status over a 24-month period using a linear mixed effects model. We modeled the outcome of protection against severe COVID-19 and infection as the log of 1 minus protective effectiveness, with predictor variables of the log of months since last vaccine dose or COVID-19 illness (whichever was more recent), age group (18-49 years, 50-64 years, 65+ years), and prior infection status, based on available literature. We modeled two immunocompromised groups, generating age-specific estimates for a mild immunocompromised group (13% lower protection) and moderate or severe immunocompromised group (25% lower protection, incorporating faster waning)^{2,16,29,30}. We assumed that each repeated booster dose would achieve the same level of effectiveness without immune exhaustion, immune imprinting phenomenon, or reduced vaccine effectiveness due to new variants^{31,32}, although we explored this in sensitivity analysis.”

Comment 7

4) The readability of the plots can be improved by making adjustments to their design. The current plots consisting of white panels and small points make it challenging for readers to interpret the data effectively. A few suggestions would be to increase the size of data points, and adding y-axis gridlines.

Response: We have modified the Figure 1 plots to increase the size of the data point and add y-axis gridlines as suggested. We have also simplified this Figure by only selecting a few risk groups, and moving the remainder to the Appendix. An example of a panel from the revised plot is shown below.

In Tables and Figures:

B. 75+ years

Comment 8

5) An additional limitation of the model is the absence of consideration for vaccine hesitancy, which can vary among different age groups and health conditions. This omission can have implications for the effectiveness of specific vaccine scenarios, so I believe it is worth at least mentioning this aspect in the Discussion.

Response: We agree with the reviewer about the relevance of vaccine hesitancy. As suggested, we have added this as a limitation in the Discussion, including a note that vaccine hesitancy is different by age group and risk factors. Finally, in our dynamic transmission model (where we model the entire US population, including unvaccinated and under vaccinated persons), we account for imperfect vaccine coverage, and hence, vaccine hesitancy (estimates in Table A6).

In Discussion:

“Finally, we did not account for vaccine hesitancy, which may vary by age group and health conditions.²³”

In References:

23. Roberts-McCarthy, E. et al. Factors associated with receipt of mRNA-1273 vaccine at a United States national retail pharmacy during the COVID-19 pandemic. *Vaccine* 41, 4257-4266 (2023).

In Methods:

“*Scenario analysis: Dynamic transmission model*

...Second, the simulated population included all age groups and unvaccinated individuals. Third, vaccine strategies were applied with imperfect uptake coverage by age- and immune status to reflect current uptake.”

Comment 9

6) It would be beneficial to have in the discussion or in the introduction a more in-depth discussion of the methodology used, mentioning if microsimulations have been used in other contexts of COVID-19, what are the pros and cons and relevant references.

Response: We appreciate the suggestion to give broader context on use of microsimulation in public health and during the pandemic. We have edited our Introduction to provide this context, and include additional references to a review article on microsimulation modeling for public health and examples from the pandemic.

In Introduction:

“Using a microsimulation model, **which is a common public health modeling approach that allows the simulation of individual people with unique characteristics**^{10,14-15}, we model severe COVID-19 to compare the individual- and population-level impact of various timings of bivalent COVID-19 booster vaccination in different risk groups.”

In References:

14. Rutter, C.M., Zaslavsky, A.M. & Feuer, E.J. Dynamic microsimulation models for health outcomes: a review. *Med Decis Making* 31, 10-8 (2011).

15. Li, Q. & Huang, Y. Optimizing global COVID-19 vaccine allocation: An agent-based computational model of 148 countries. *PLoS Comput Biol* 18, e1010463 (2022).

Comment 10

7) The abstract reads “annual and semiannual bivalent boosters would reduce annual absolute risk of severe COVID-19 by 311 (277-369) and 578 (494-671) cases, respectively, compared to a one-time bivalent booster dose.”, I think these numbers are per 100,000 individuals, it would be more clear to mention this explicitly.

Response: We have revised this sentence to state the estimate is per 100,000 persons.

“Analyzing United States COVID-19 surveillance and seroprevalence data in a microsimulation model, we estimated that in persons 75+ years, annual and semiannual bivalent boosters would reduce annual absolute risk of severe COVID-19 by 199 (uncertainty interval 188-229) and 368 (344-413) cases **per 100,000 persons**, respectively, compared to a one-time bivalent booster dose.”

Comment 11

8) Page 14: “We estimated this risk for persons vaccinated with the primary series and at least one monovalent booster This calibration yielded...” I think a “.” is missing between “booster” and “This”.

Response: We have corrected this.

Response to Reviewer 2:

General comment

The authors present a very clear and well-structured analysis of the respective benefits of different vaccination frequencies in preventing severe COVID-19 disease amongst different cohorts in the US population. The results are based on current vaccines in use in the US against Omicron type infection characteristics collated from a variety of existing sources and applied using a simple model for immunity from disease at an individual level.

Response: We appreciate the helpful and constructive comments.

Comment 1

1) There are however a number of limitations to the approach that affect the use of the results in providing a reliable future forecast. Firstly, vaccine effects are only considered at a personal and not population level. Since the study only considers efficacy versus severe disease effects, and not in limiting susceptibility/transmissibility/symptom prevalence, the use of vaccines in limiting infection spread is not considered— something which was considered to be very important when vaccines for COVID were first introduced. This is perhaps okay in a situation where future vaccination is concentrated only on protecting the vulnerable, but certainly needs more discussion.

Response: We appreciate the reviewer’s comment on the importance of considering transmission dynamics in the analysis by accounting for the indirect effects of vaccination. We agree with the reviewer about considering indirect effects of vaccination and have therefore **added a new analysis using a dynamic transmission model**, which accounts for the indirect effects of vaccination on transmission dynamics. We include this as a separate and complementary analysis to the original model (see new Figure 3). Specifically, we compare frequent booster vaccine strategies focused only on the highest risk persons (75+ years, moderate/severe immunocompromised group) versus a more inclusive strategy (18+ years in all groups), expecting that indirect effects would be more pronounced with a more inclusive strategy. This analysis is designed to test the importance of considering transmission dynamics in the analysis.

While the dynamic model demonstrates some modest differences in impact from indirect effects attributable to more frequent booster vaccination with a more inclusive strategy (18+ in all age groups) compared to a targeted strategy (75+ years, severely immunocompromised only), within the assumed conditions, the overall model findings are broadly similar. This is likely since: i) booster vaccines have indirect protection, but they are relatively short-lived and modest; ii) vaccine booster uptake is likely to be low within the population based on current observed coverage; iii) a substantial proportion of the population remains unvaccinated or under vaccinated. Therefore, under realistic assumptions, the indirect effects of booster vaccination are likely to be modest given low uptake of vaccines and modest indirect effects of booster vaccines. These conditions all favor that

transmission dynamics are unlikely to change our study's overall conclusions. We have included the new analysis in the study and additional new text to the Discussion section to describe these findings.

We have kept the static model as the main analysis for several reasons. First, as we demonstrate with the new sensitivity analysis using a dynamic transmission model, the static model is a valid modeling approach to answer the study question. Second, the static model allows us to better calibrate to observed data on severe COVID-19 cases, seroprevalence, and other key parameters, which are challenging to do with a dynamic model. Third, the dynamic transmission model forces our model to make many additional assumptions that have limited data, such as age-based mixing and transmission amongst others. Fourth, the public health and clinical question of our study focuses on vaccinated populations and adults; however, the dynamic transmission model requires modeling children and unvaccinated persons and anticipated coverage, which are uncertain. Therefore, we have kept the original model as the main analysis but do include the new dynamic transmission model suggested by the reviewer.

The methodological details and results of the new analysis using a dynamic transmission model are outlined. We also add additional Discussion about this important consideration. Our new results broadly support a similar conclusion.

Please see Reviewer 1, Comment 1 for a complete list of changes in this revised manuscript, including new Methods, Results, Appendix Methods, and the new Figure 3.

Comment 2

2) Secondly, due to continued high levels of infection, COVID-19 is still rapidly evolving and immunity from both previous infection and current vaccines is highly volatile. It is likely that vaccines will continued to be adapted to meet these challenges (in a similar manner to seasonal influenza), but COVID has yet to settle into endemicity, leaving high levels of uncertainty. While the authors do a good job in presenting parameter sensitivity, future disease could easily be expected to fall outside of their predictions.

Response: We appreciate the reviewer's comment on the importance of considering further potential scenarios related to SARS-CoV-2 evolution and immune evasion from vaccines and natural infection. Beyond the sensitivity analyses we have done, we have **added new scenario analyses that simulate different scenarios of novel variants with immune evasion**. We simulate emergence of a new variant (reduced susceptibility to vaccine and natural protection) at the start of the simulation (scenario 1), and at 12 months (scenario 2). We simulate new variants annually (scenario 3), and also use of a seasonally targeted vaccine (scenario 4). This has been added as a new Figure 2 in the main text, with additional description of this new analysis described below.

In Methods:

“Scenario analysis: Simulation of novel variants

We repeated the primary analysis under different scenarios for emergence of novel variants with immune evasion (Figure 2A), including one scenario with a variant targeted vaccine. Upon circulation of a novel variant, we modeled two different scenarios: i) absolute protection from vaccine or hybrid protection against non-severe and severe COVID-19 is reduced by 10%, due to immune evasion; and ii) absolute protection is reduced by 10%, and rate of waning increases by 5%. We did not simulate variants with higher infectiousness or

severity. In the scenario with a variant targeted vaccine, we assumed the vaccine restored the protection lost due to the new variant. Novel variants were introduced over a 3-month period. A full description of the analysis is available in the Appendix.”

In Appendix:

“Scenario analysis: Novel variants

We repeated the primary analysis under different scenarios for emergence of novel variants with immune evasion (summarized in Figure 2A). In scenario 1, a novel variant is introduced at the start of the simulation. In scenario 2, a novel variant is introduced at the start of Year 2 of the simulation. In scenario 3, a novel variant 1 is introduced at the start of the simulation, and a novel variant 2 is introduced at the start of Year 2. In scenario 4, the novel variant circulation is the same as outlined in scenario 3, but this time, the vaccines administered are targeted to the variant (two distinct vaccine formulations, akin to the seasonally targeted influenza vaccine). Novel variants were introduced over a 3-month period. Variants are modeled under two different immune evasion scenarios: i) absolute protection from vaccine or hybrid protection against severe COVID-19 is reduced by 10% with circulation of the novel variant, due to immune evasion; and ii) absolute protection is reduced by 10%, and rate of waning increases by 5% with circulation of the novel variant. We therefore simulated 8 total scenarios, with 4 variant scenarios and 2 immune evasion scenarios. We did not simulate variants with higher infectiousness or severity. In the scenario with a variant targeted vaccine, we assumed the vaccine restored the protection lost due to the new variant.”

In Results:

“We repeated the primary analysis under different scenarios with emergence of novel variants with immune evasion (summarized in Figure 2A). Scenarios simulating novel variants with immune evasion increased overall number of severe COVID-19 cases, although the overall impact of more frequent booster vaccines by risk group was similar. In those 65-74 years old, semiannual booster vaccination under annual novel variant circulation (scenario 3) would lead to an annual risk reduction of 138 severe cases per 100,000 persons, whereas under no novel variant introduction (primary analysis) would lead to an annual risk reduction of 142 severe cases per 100,000 persons, compared to a one-time booster vaccination. The scenario with a variant targeted vaccine had larger benefits of more frequent booster vaccines. In persons 75+ years old, semiannual booster vaccination with a variant targeted vaccine (scenario 4) would lead to an annual risk reduction of 190 severe cases per 100,000 persons, whereas with a non-targeted vaccine (scenario 3) would lead to an annual risk reduction of 138 severe cases per 100,000 persons compared to a one-time booster vaccination.”

In Discussion:

“Decisions on frequency of COVID-19 booster vaccination are likely to be influenced by emergence of novel variants and new formulations of vaccine. In this study, we simulated different potential scenarios for emergence of novel variants with immune evasion, although the full range of evolutionary possibilities for variant characteristics (e.g., infectiousness, severity of illness, mechanism of immune evasion) are difficult to capture. We simulated variant scenarios with evasion of protection generated by vaccine and hybrid immunity and found our overall study findings to be robust, with larger benefit with variant-targeted vaccines.”

“These findings were similar when accounting for indirect effects of vaccines on transmission, and emergence of novel variants with immune evasion.”

In Tables and Figures:
“

A. Variant Scenarios Explanation

B. 18-49 years

C. 65-74 years

D. Immunocompromised (Mild)

Figure 2: Scenario analysis on emergence of novel SARS-CoV-2 variants comparing severe COVID-19 risk with different frequencies of COVID-19 booster vaccination. We simulated

four scenarios on emergence of novel variant(s) with reduced susceptibility to protection generated by prior vaccination and natural infection (panel A). Under each variant scenario analysis, we simulated three frequencies of COVID-19 booster vaccine for each key group. Additional variant scenarios and risk groups available in the Appendix. We plotted absolute annual risk of severe COVID-19 over a two-year simulation. The vertical bars represent uncertainty intervals, which simulate different scenarios of baseline conditions to account for uncertain model inputs. Intervals are designed to demonstrate uncertainty within a single vaccine strategy rather than for comparison between vaccine strategies; vaccine strategies should be compared using the same assumed baseline conditions.”

Comment 3

3) Finally, the study is inherently US specific. Vaccine use (type and prevalence), healthcare, and demographics vary greatly across the world, all of which greatly affect the occurrence of severe disease. Data sources and fitting are exclusively US specific, and I suggest the authors should consider adjusting the title to reflect this.

Response: We agree with the reviewer and have revised the Title as suggested (shown below).

In Title:

“**Comparison of timing of booster vaccination for COVID-19 to prevent severe disease by risk group in the United States**”

Response to Reviewer 3:

General comment

This manuscript estimates the health benefit (in terms of averted cases of severe COVID-19) of different frequencies of booster vaccination for different population groups. I have some questions about the set-up of the scenarios that are investigated in the paper, and some smaller comments.

Response: We appreciate the helpful and constructive comments.

MAJOR COMMENT

Comment 1

1) For the analysis of immunocompromised individuals, I worry that looking at a single group simplified the analysis too much. As the extent of immune deficiency will differ between health conditions, I would expect the benefits of vaccination would also differ condition by condition. can this be reflected in the paper (preferably in analysis or at least in discussion)?

Response: We agree with the reviewer’s point on the importance of modeling multiple immunocompromised groups given the wide range of vaccine responses and immunity/vaccine-induced protection expected within this population. We have revised the main analysis to include two immunocompromised groups (mild vs moderate/severe) that are each age-stratified (new Tables S3 and S4). Furthermore, we assume different dynamics of vaccine response between the mild vs moderate/severe immunocompromised groups, as described below, to better evaluate different scenarios. While we provide aggregated results from this new analysis (mild vs moderate/severe

immunocompromised group) in the main text, we include the full age-stratified results in the Appendix.

In Methods:

“We modeled two immunocompromised groups, generating age-specific estimates for a mild immunocompromised group (e.g., low-dose corticosteroids, mild immunosuppressive medications; 13% lower protection) and moderate or severe immunocompromised group¹⁹ (e.g., hematologic malignancy with active treatment or poor response to vaccines, solid organ or bone marrow transplant, high-dose corticosteroids or other moderate/severely immunosuppressive medications¹⁹; 25% lower protection, incorporating faster waning).^{2,19}”

In Results:

“In a hypothetical cohort of mild and moderate/severe immunocompromised persons who received a one-time booster vaccination, the model estimated an annual risk of 1,290 (UI: 1,205-1,403) and 1,367 (UI: 1,266-1,503) cases per 100,000 persons, respectively. For mild immunocompromised persons, annual and semiannual booster vaccination reduced absolute annual risk by 110 (UI: 87-117) and 195 (UI: 148-217) cases per 100,000 persons respectively, compared to one-time booster. For moderate/severe immunocompromised persons, annual and semiannual booster vaccination reduced absolute annual risk by 184 (UI: 175-196) and 310 (UI: 300-320) cases per 100,000 persons respectively, compared to one-time booster.”

In Discussion:

“We include two distinct immunocompromised groups, although acknowledge there is heterogeneity within these populations that is not captured.”

In Tables and Figures:

Table 1: Number of severe COVID-19 cases, risk, and number needed to treat to avert severe COVID-19 in six risk groups with different frequencies of COVID-19 booster vaccination.

	Total severe COVID-19 cases ^a	Absolute annual risk of severe COVID-19 (cases per 100,000; 95% UI)	Annual risk reduction of severe COVID-19		% Averted severe COVID-19		NNT to avert severe COVID-19 case ^a
			Absolute risk (cases per 100,000)	Relative risk (%)	No Prior Infection ^b	Prior Infection ^b	
One-time booster ^c							
18-49 years	1,954	98 (85-125)	--	--	--	--	--
50-64 years	3,978	199 (185-238)	--	--	--	--	--
65-74 years	10,484	524 (499-562)	--	--	--	--	--
75+ years	27,955	1,398 (1,332-1,501)	--	--	--	--	--
Immunocompromised (Mild) ^d	25,805	1,290 (1,205-1,403)	--	--	--	--	--

Immunocompromised (Moderate/Severe) ^d	27,343	1,367 (1,266-1,503)	--	--	--	--	--
Annual booster							
18-49 years	1,671	84 (74-106)	14	14%	48%	52%	3,534
50-64 years	3,424	171 (159-202)	28	14%	68%	32%	1,806
65-74 years	8,924	446 (425-475)	78	15%	83%	17%	648
75+ years	23,966	1,198 (1,144-1,272)	199	15%	83%	17%	251
Immunocompromised (Mild) ^d	23,609	1,180 (1,088-1,316)	110	9%	67%	33%	456
Immunocompromised (Moderate/Severe) ^d	23,669	1,183 (1,091-1,307)	184	13%	50%	50%	273
Semiannual booster (every 6 months)							
18-49 years	1,432	72 (64-90)	26	27%	46%	54%	1,916
50-64 years	2,944	147 (136-171)	52	26%	67%	33%	968
65-74 years	7,645	382 (365-404)	142	27%	83%	17%	353
75+ years	20,602	1,031 (988-1,088)	368	26%	82%	18%	136
Immunocompromised (Mild) ^d	21,899	1,095 (988-1,255)	195	15%	67%	33%	257
Immunocompromised (Moderate/Severe) ^d	21,138	1,057 (966-1,183)	310	23%	51%	49%	162

^aEstimated over 2-year simulation period in population of 1 million persons for each risk group.

^bPrior infection status based on start of simulation.

^cOne-time booster is the baseline intervention for risk reduction calculations.

^d **Definitions for each immunocompromised status are available in the Methods. We report age-weighted estimates in this Table. Full age-stratified results for the immunocompromised population is available in the Appendix.**

NNT; number needed to treat, which is based on the number of persons (instead of vaccine doses) needing to follow a vaccine schedule to avert one severe COVID-19 case

Scenario with no booster is available in Table S2.

In Appendix:

Table S3: Number of severe COVID-19 cases, risk, and number needed to treat to avert severe COVID-19 in four age groups among the **mild immunocompromised population** with different frequencies of bivalent COVID-19 booster vaccination.

	Total severe COVID-19 cases ^a	Absolute annual risk of severe COVID-19 (cases per 100,000; 95% UI)	Annual risk reduction of severe COVID-19		% Averted severe COVID-19		NNT to avert severe COVID-19 case ^a
			Absolute risk (cases per 100,000)	Relative risk (%)	No Prior Infection	Prior Infection	

One-time booster ^b							
18-49 years	5,851	293 (267-339)	--	--	--	--	--
50-64 years	11,746	587 (539-668)	--	--	--	--	--
65-74 years	30,065	1,503 (1,411-1,612)	--	--	--	--	--
75+ years	80,148	4,007 (3,763-4,296)	--	--	--	--	--
Annual booster							
18-49 years	5,441	272 (243-322)	21	7%	48%	52%	2,440
50-64 years	10,863	543 (490-627)	44	8%	73%	27%	1,133
65-74 years	27,311	1,377 (1,270-1,492)	138	9%	83%	17%	364
75+ years	73,147	3,657 (3,391-4,033)	350	9%	82%	18%	143
Semiannual booster (every 6 months)							
18-49 years	5,173	258 (224-309)	34	12%	49%	51%	1,475
50-64 years	10,070	504 (445-602)	84	14%	71%	29%	597
65-74 years	25,180	1,259 (1,148-1,421)	244	16%	83%	17%	205
75+ years	67,745	3,387 (3,076-3,841)	620	15%	82%	18%	81

^aEstimated over 2-year simulation period in population of 1 million persons.

^bOne-time bivalent booster is the baseline intervention for risk reduction calculations.

NNT; number needed to treat, which is based on the number of persons needing to follow a vaccine schedule to avert one severe COVID-19 case

Table S4: Number of severe COVID-19 cases, risk, and number needed to treat to avert severe COVID-19 in four age groups among the **moderate/severe immunocompromised population** with different frequencies of bivalent COVID-19 booster vaccination.

	Total severe COVID-19 cases ^a	Absolute annual risk of severe COVID-19 (cases per 100,000; 95% UI)	Annual risk reduction of severe COVID-19		% Averted severe COVID-19		NNT to avert severe COVID-19 case ^a
			Absolute risk (cases per 100,000)	Relative risk (%)	No Prior Infection	Prior Infection	
One-time booster ^b							
18-49 years	6,153	308 (278-353)	--	--	--	--	--
50-64 years	12,498	625 (570-715)	--	--	--	--	--
65-74 years	31,891	1,595 (1,488-1,724)	--	--	--	--	--
75+ years	84,909	4,245 (3,947-4,624)	--	--	--	--	--

Annual booster							
18-49 years	5353	268 (243-308)	40	13%	31%	69%	1,250
50-64 years	10,829	541 (491-622)	83	13%	51%	49%	600
65-74 years	27,471	1,374 (1,276-1,489)	221	14%	69%	31%	227
75+ years	73,542	3,677 (3,402-4,024)	568	13%	68%	32%	88
Semiannual booster (every 6 months)							
18-49 years	4,842	242 (215-280)	66	21%	32%	68%	763
50-64 years	9,649	482 (435-565)	142	23%	52%	48%	352
65-74 years	24,382	1,219 (1,124-1,345)	375	24%	71%	29%	134
75+ years	65,716	3,286 (3,017-3,642)	960	23%	69%	31%	53

^aEstimated over 2-year simulation period in population of 1 million persons.

^bOne-time bivalent booster is the baseline intervention for risk reduction calculations.

NNT; number needed to treat, which is based on the number of persons needing to follow a vaccine schedule to avert one severe COVID-19 case

Comment 2

2) The results are presented as providing information on the individual benefits of vaccination, but do not appear to stratify results by information that an individual presumably would have – that is, the recency of prior vaccination and infection. As this information would impact an individual's existing level of immunity, shouldn't this be considered with individuals make vaccination decisions? Response: We appreciate the reviewer's point that vaccine decisions may be influenced by baseline risk and projected benefit, both of which are influenced by time since last vaccine or infection, which are often known to an individual. The goal of this study was to provide general conclusions to inform public health guidance, rather than patient level decisions, although we acknowledge this data can be useful for these decisions as well. To address the reviewer's point, we include a new set of Figures that account for baseline risk and waning protection over time for personalized individual decisions, which are available in the Appendix.

In Appendix:

A. 18-49 years

C. 65-74 years

B. 50-64 years

D. 75+ years

Figure S2: Risk of severe COVID-19 over time by baseline risk and waning protection.

We modeled risk of severe COVID-19 by baseline risk and time since last immune event (vaccine or infection) by multiplying age-specific lambdas to protection estimates over time. The risk groups modeled here are the (A) 18-49 years; (B) 50-64 years; (C) 65-74 years; (D) 75+ years; (E) Immunocompromised (mild); and (F) Immunocompromised (moderate/severe).

Comment 3

3) Why was the immunocompromised group not stratified by age? Naively, one might conclude a 75-year-old with HIV infection (220 per 100,000) is at lower risk than a 75-year-old without HIV infection (311 per 100,000). Clearly this is not the intended interpretation, but the current results format makes it hard to understand risk for immunocompromised individuals at different ages, yet it is known that age is a big risk factor.

Response: We agree with the reviewer and have now conducted the analysis of immunocompromised groups as an age-stratified analysis. This is further described in Comment #1. This comment highlights changes in the Methods, Results, Table. Additional new Tables (Tables S2-S3) are generated that provide age-stratified estimates for the two immunocompromised groups.

Comment 4

4) I am surprised that the sensitivity analyses presented in the main text do not include the introduction of a new variant with substantial immune escape – would this have an effect on the results? On page 13 it is mentioned that this is explored in sensitivity analyses, but these results are not seen in Figure 1.

Response: We appreciate the reviewer’s comment on the importance of considering new scenarios of viral evolution, such as emergence of new SARS-CoV-2 variants with different levels of immunity against prior infection and current vaccines. To address this important point, we have **added four new analyses that simulate different scenarios of novel variants with immune evasion**. We simulate emergence of a new variant (reduced susceptibility to vaccine and natural protection) at the start of the simulation (scenario 1), and at 12 months (scenario 2). We simulate new variants annually (scenario 3), and also use of a seasonally targeted vaccine (scenario 4). This has been added as a new Figure 2 in the main text, with additional description of this new analysis.

Please see Reviewer 2, Comment 2 for a complete list of changes in this revised manuscript, including Methods, new Results, and a new Figure 2 for novel variant scenario analysis.

Comment 5

MINOR COMMENTS

1) P2: I suggest adding “per 100,000 persons” for the first set of quantitative results reported (“reduce annual absolute risk of severe COVID-19 by 311 (277-369) and 578 (494-671) cases”), otherwise it seems like these are results for the whole US population, not per 100,000.

Response: We have revised this sentence to state the estimate is per 100,000 persons.

“Analyzing United States COVID-19 surveillance and seroprevalence data in a microsimulation model, we estimated that in persons 75+ years, annual and semiannual bivalent boosters would reduce annual absolute risk of severe COVID-19 by 199 (uncertainty interval 188-229) and 368 (344-413) cases **per 100,000 persons**, respectively, compared to a one-time bivalent booster dose.”

Comment 6

2) P4, paragraph 3: The following is confusing to me: “the model estimated 2,158 severe COVID-19 cases over a 2-year period in a population of 1 million people with annual risk ” It seems like this is basically the same statistic, just reported in different units?

Response: The reviewer is correct. We have simplified the reporting to improve clarity.

In Results:

“In a hypothetical cohort of persons 18-49 years old who received a one-time bivalent booster vaccination, the model estimated ~~2,158 severe COVID-19 cases over a 2-year period in a population of 1 million people with~~ an annual risk of 108 severe cases per 100,000 persons (95% UI: 105-108).”

Comment 7

3) P5, paragraph 1: “without prior documented COVID-19 infection” would clarify whether this was from the start of the simulation, or at the point of developing severe disease (for example, consider someone infected in year 1, and then again in year 2, and developed severe disease in year 2).

Response: This refers to the start of the simulation, and we have clarified this in the text and footnote for Table 1.

In Results:

“In the annual and semiannual vaccine strategy, we estimated that 36% and 40% of the averted severe COVID-19 cases occurred in persons without prior documented COVID-19 infection **at the start of the simulation**, respectively.”

Comment 8

4) P6, paragraph 3: “Additional results for sensitivity analysis are available in the Appendix.” I would suggest summarizing these results in the main text, even if this needs to be done succinctly.

Response: We appreciate the opportunity to clarify. This statement was meant to describe that the full results/tables for the already described sensitivity analyses are available in the Appendix. We have clarified this. We have also re-reviewed this section to update any additional sensitivity analyses.

In Results:

“**Full results for each sensitivity analysis are available in the Appendix.**”

Comment 9

5) P7, paragraph 1: For the statement “We found ... than a single recommendation for the general population” I suggest rewording this – as this was not a comparison that was done in the analysis I am not sure this should be presented as a conclusion of the study as it currently is.

Response: We have removed this comparison as suggested.

In Discussion:

“We found that more frequent COVID-19 booster vaccination in older populations and those with immunocompromising conditions at risk for severe COVID-19, along with less frequent booster vaccination in younger, low-risk populations may more effectively mitigate the burden of severe COVID-19 in the United States ~~than a single recommendation for the general population.~~”

Comment 10

6) P8, paragraph 1: A picky point: for the text “influenza vaccination to prevent death (NNT 48)” I worry about providing this number for comparison, as it seems an extremely low estimate of the NNT for this intervention and outcome (e.g. would need vaccination to reduce flu risk from 100% to 0%, and for flu to have an IFR of 2%, or some similar combination of assumptions). I realize that this NNT comes from a cited study, but still it stands out.

Response: We agree with the reviewer and have removed the NNT estimate from this example.

“For example, common primary care measures have a range of NNT from 50-400, such as influenza vaccination to prevent death (~~NNT 48~~), statin for primary prevention of death (NNT 286), and colonoscopy to prevent colon cancer associated death (NNT 445)¹⁷⁻¹⁹”

Comment 11

7) Figure A1 is so central to the analysis it would be great if there were room for it in the main text.

Response: We agree with the reviewer. In this revision, we have added two new Figures (with multiple panels) for the new variant analysis and dynamic model. If the Editor allows, we can also move an abbreviated version of these figures to the main text.

Comment 12

8) Table A2: the table note reads “stochastic variation explains the discrepancy between model and observed output”. Can you include intervals for the model output to confirm this? For 75+ and Immunocompromised groups the difference in incidence is a bit concerning.

Response: We have updated this Table and clarified as suggested, demonstrating an accurate calibration. The Table A2 is now corrected.

Comment 13

9) Figure A4: apologies if I am understanding the quantities being plotted – is the line the central estimate (eg, mean, or median), and are the shaded areas the interval? If so, why does the central estimate fall outside the interval in some situations? I have the same concern about Figure S1. Also, some of the intervals in the tables appear to suffer the same problem (from Table S4, page 18, we see intervals 1,337 (1,321 – 1,337) and 193 (176 – 194), which don’t seem right.

Response: Table A4 and Figure S1 have been updated to resolve this.

REVIEWER COMMENTS

Reviewer #1 (Remarks to the Author):

I would like to thank the authors for addressing all of my points and for developing an additional analysis with a dynamic transmission model. I think the paper has significantly improved with this addition, and the solidity of the results has improved as well.

Therefore, I recommend this paper for publication in Nature Communications.

Reviewer #2 (Remarks to the Author):

I would like to thank the authors for taking my original comments on board. The revised manuscript appears greatly improved and goes some way to addressing my original concerns. I do however have a few remaining comments and queries on how this has been done.

In the section for the “dynamic transmission model”, I’m worried that the methodology being used is insufficient to support the results. It is stated “we assumed equal mixing between age groups and homogenous mixing overall”. Mixing tends to be much greater for the younger population, and subsequently indirect vaccine effects (ie. on transmission) are expected to be much greater on these groups than the elderly. To understand how indirect effects of vaccination may affect vaccine group targeting strategies, it is therefore essential that inhomogeneous mixing is accounted for in some way.

In the section for the “simulation of novel variants”, it appears to me that (Figure 2) Scenario 1 and 3 are apparently identical? I’m surprised by this, can I check does variant 2 give an extra 10% reduction beyond variant 1— so 20% reduction in immunity resulting from vaccination and infection with original variant, and 10% reduction in immunity due to variant 1? This would be the natural approach I think (unless you can convincingly argue otherwise), since antigenic distance should expect to increase over time. It should also be noted that the variant scenarios considered are extremely conservative. Novel variants, causing new waves of infection, are currently coming being seen with great frequency— monthly rather than yearly (see the GISAID database for instance).

If these final points can be addressed, I would be happy to support publication.

Reviewer #3 (Remarks to the Author):

Thank you for these revisions and explanations. I am largely satisfied with these changes, with the following remaining points (apologies, the first two are new).

- In the text I see the term “uncertainty interval”, but would specify that these are 95% intervals, if that is the case (also in table footnotes). And in methods could note whether these are equal-tailed or HPD intervals. Given the question below, the authors might also describe somewhere how the point estimates are calculated (mean of simulations, median of simulations, model evaluated using parameter means, etc), if not done already.
- Though I am not in a position to confirm whether this is actually an issue, the comparison of point estimates and interval bounds in Figure 2 B & C is surprising to me. In these plots, the point estimate appears much closer to the lower bound than the upper bound (in some cases, point estimate minus lower bound appears $\frac{1}{4}$ the value of upper bound minus point estimate). In Figure 3 the point estimates are more modestly shifted towards the top of the interval, so I suspect this is not an inherent feature of the quantities being modelled. Is the point estimate calculated as the

mean or median of the simulation results used to construct the interval? If not (e.g., if calculated from parameter point estimates) that could potentially explain this issue. The interval could also look strange if constructed from a very small number of parameter sets. Anyway, I would suggest tracking down why this is happening to confirm that there is no problem with the intervals or point estimates presented.

- For the comment from the first review on the use of an NNT of 48: in the revision the number 48 is removed, but the text still states "common primary care measures have a range of NNT from 50-400". If the NNT of 48 for influenza is what justifies the 50 in "50-400", then I think the issue still remains.

Park *et al.* “Comparison of timing of booster vaccination for COVID-19 to prevent severe disease by risk group in the United States”

(Reference no. NCOMMS-23-29382A) – Point-by-point response

Response to Reviewer 1:

General comment

I would like to thank the authors for addressing all of my points and for developing an additional analysis with a dynamic transmission model. I think the paper has significantly improved with this addition, and the solidity of the results has improved as well.

Therefore, I recommend this paper for publication in Nature Communications.

Response: Thank you for your review of our study.

Response to Reviewer 2:

General comment

I would like to thank the authors for taking my original comments on board. The revised manuscript appears greatly improved and goes some way to addressing my original concerns. I do however have a few remaining comments and queries on how this has been done.

Response: Thank you for these additional comments, which we address below.

Comment 1

In the section for the “dynamic transmission model”, I’m worried that the methodology being used is insufficient to support the results. It is stated “we assumed equal mixing between age groups and homogenous mixing overall”. Mixing tends to be much greater for the younger population, and subsequently indirect vaccine effects (ie. on transmission) are expected to be much greater on these groups than the elderly. To understand how indirect effects of vaccination may affect vaccine group targeting strategies, it is therefore essential that inhomogeneous mixing is accounted for in some way.

Response: We thank the reviewer for the comment on the contact matrix incorporated in the dynamic transmission model. We have revised the primary dynamic model to include heterogenous social mixing by age group, using published contact matrices between age groups (Prem et al., PLOS Comp Bio 2017; Mossong et al., PLOS Med 2008). These contact matrices (shown below) account for differences in age group mixing as noted by the reviewer. While there is certainly variation and uncertainty in these contact structures, we believe this is the best available data to address this comment.

We find that the results remain overall similar. The Methods section and Results have all been revised accordingly.

In Appendix:

“We applied an age-based contact matrix to account for heterogeneous mixing by age group, using term $C_{j,k}$ to account for the number of contacts C between an individual of age group j with another age group k in the United States (see Table A6)²⁰.”

Table A6. Age-based contact matrix for dynamic transmission model.

Contacts (per day) ^a	0-17 years	8.35	2.88	0.83	0.31	0.14
	18-49 years	5.55	9.98	3.22	0.48	0.23
	50-64 years	1.99	2.96	2.93	0.56	0.21
	65-74 years	0.61	0.73	0.79	1.17	0.31
	75+ years	0.31	0.27	0.26	0.24	0.40
		0-17 years	18-49 years	50-64 years	65-74 years	75+ years
	Individual					

^aThese are average contacts per day. We adjust these for a week time step.

In Methods (Scenario analysis: Dynamic transmission model):

“First, the ‘force of infection’ term was formulated to be directly related to the number of SARS-CoV-2 infections in the population in the prior time step (week) **with age-specific contact matrices.**³⁵⁻³⁶”

References:

35. Mossong J, Hens N, Jit M, et al. Social Contacts and Mixing Patterns Relevant to the Spread of Infectious Diseases. *PLOS Med.* 2008;5(3):e74. doi:10.1371/journal.pmed.0050074

36. Prem K, Cook AR, Jit M. Projecting social contact matrices in 152 countries using contact surveys and demographic data. *PLOS Comput Biol.* 2017;13(9):e1005697. doi:10.1371/journal.pcbi.1005697

In Results:

“To investigate the impact of indirect effects of vaccination on transmission, we repeated the primary analysis using a dynamic transmission model (Figure 3). We found that indirect effects were larger with more inclusive frequent booster vaccine strategies, although within the assumed conditions and realistic vaccine uptake, the overall model conclusions were broadly similar to the primary (static) model. **In a focused vaccination program in high-risk populations (75+ years and moderate/severe immunocompromised groups) under realistic vaccine coverage assumptions, the dynamic model estimated that annual and semiannual booster vaccination would lead to an annual risk reduction of 209 (UI: 186 – 258) and 450 (UI: 387 – 518) severe cases per 100,000 persons in those 75+ years, compared to a one-time booster vaccination (Figure 3A). In a more inclusive vaccination program (18+ years all groups), the model**

estimated that annual and semiannual booster vaccination would lead to an annual risk reduction of 257 (UI: 229 – 295) and 602 (UI: 513 – 683) severe cases per 100,000 persons in those 75+ years, compared to a one-time booster vaccination (Figure 3A). Under more optimistic vaccine coverage assumptions, indirect effects were larger (Figure 3B).”

In Figures/Tables:

A. Realistic vaccine coverage

B. Optimistic vaccine coverage

Figure 3: Scenario analysis using a dynamic transmission model to estimate the impact of indirect effects on COVID-19 booster vaccination strategies in the 75 years and older group.

We used a dynamic transmission model to estimate the impact of different frequencies of COVID-19 booster vaccination across different groups would affect transmission in the highest risk populations (75+ years). We simulated booster vaccination with varying levels of inclusiveness: (i) 18+ years in all groups (most inclusive); (ii) 65+ years and all immunocompromised groups; and (iii) 75+ years, moderate/severe immunocompromised group (most restrictive). We simulated under realistic vaccine coverage (panel A) and optimistic coverage (panel B) assumptions. We assumed a background of one-time booster vaccination at the start of the simulation in adults (18+ years) with age-specific, imperfect vaccine uptake. We plotted absolute annual risk of severe COVID-19 over a two-year simulation in the 75+ year risk group, to compare the indirect effects of booster vaccination on this high-risk group. The largest indirect effects from vaccination are expected with more inclusive vaccine strategies. The vertical bars represent uncertainty intervals and capture the

full range of varied model parameters, while the point estimate uses base case assumptions of model inputs. Intervals are designed to demonstrate uncertainty within a single vaccine strategy; comparison between vaccine strategies should be use the same assumed baseline conditions. A full description of the Methods and results for additional risk groups are available in the Appendix.

Comment 2

In the section for the “simulation of novel variants”, it appears to me that (Figure 2) Scenario 1 and 3 are apparently identical? I’m surprised by this, can I check does variant 2 give an extra 10% reduction beyond variant 1— so 20% reduction in immunity resulting from vaccination and infection with original variant, and 10% reduction in immunity due to variant 1? This would be the natural approach I think (unless you can convincingly argue otherwise), since antigenic distance should expect to increase over time. It should also be noted that the variant scenarios considered are extremely conservative. Novel variants, causing new waves of infection, are currently coming being seen with great frequency— monthly rather than yearly (see the GISAID database for instance).

If these final points can be addressed, I would be happy to support publication.

Response: We thank the reviewer for asking about this clarification and suggestion for additional analysis. First, we would like to clarify the original variant model. In our original model, scenarios 1 and 3 are different. In these scenarios, prior infection with a homologous variant to what is currently circulating confers the highest level of protection (i.e., no reduction in protection), while prior infection with a heterologous variant to what is currently circulating confers lower protection (i.e., 10% reduction in protection). However, we did not incorporate an additional 10% reduction to account for increasing antigenic distance between variant 2 and 3, compared to variant 1. To address the reviewer’s point, we have further revised this model to add an additional 10% reduction as suggested (i.e., during variant 2 circulation, persons with infection with the original variant experience a 20% reduction in protection, while those with infection with variant 1 experience a 10% reduction in protection, compared to the original protection curves). To clarify the methods for the reader, we have also added additional Figures for each variant scenario (new Figures A8-A9). Finally, we would like to highlight that while novel variants are more frequent, those with empirical data to support significant immune evasion are less frequent; our analysis aims to address this important epidemiologic phenomenon but there is limited data to inform highly complex variant analyses at this point. We have revised the Methods section, updated all Results, and clarified this limitation as shown below.

In Appendix:

“In scenario 3 and 4, emergence of variant 2 led to an additional reduction in absolute protection in the population, beyond the initial reduction experienced during emergence of variant 1.”

See new Figure A8 and A9.

In Results:

“Scenarios simulating novel variants with immune evasion increased overall number of severe COVID-19 cases, although the overall impact of more frequent booster vaccines by risk group was similar; uncertainty in this analysis was larger. In those 65-74 years old, annual and semiannual booster vaccination under annual novel variant circulation (scenario 3) would lead to an annual risk reduction of 73 (UI: 68 – 76) and 134 (UI: 123 – 135) severe cases per 100,000

persons, respectively, compared to a one-time booster vaccination. Under the primary analysis (without novel variant introduction) this would lead to an annual risk reduction of 78 and 142 severe cases per 100,000 persons. The scenario with a variant-targeted vaccine had larger benefits of more frequent booster vaccines. In persons 65-74 years old, annual and semiannual booster vaccination with a variant-targeted vaccine (scenario 4) would lead to an annual risk reduction of 130 (UI: 120 – 146) and 233 (UI: 204 – 248) severe cases per 100,000 persons, respectively, compared to a one-time booster vaccination.”

In Tables/Figures:

A. Variant Scenarios Explanation

B. 18-49 years

C. 65-74 years

D. Immunocompromised (Mild)

Figure 2: Scenario analysis on emergence of novel SARS-CoV-2 variants comparing severe COVID-19 risk with different frequencies of COVID-19 booster vaccination. We simulated four scenarios on emergence of novel variant(s) with reduced susceptibility to protection generated by prior vaccination and natural infection (panel A). Under each variant scenario analysis, we simulated three frequencies of COVID-19 booster vaccine for each key group. Additional variant scenarios and risk groups available in the Appendix. We plotted absolute annual risk of severe COVID-19 over a two-year simulation. The vertical bars represent uncertainty intervals and capture the full range of varied model parameters, while the point estimate uses base case assumptions of model inputs. Intervals are designed to demonstrate uncertainty within a single vaccine strategy; comparison between vaccine strategies should be use the same assumed baseline conditions.

In Discussion:

“In this study, we simulated different potential scenarios for emergence of novel variants with immune evasion, although the full range of evolutionary possibilities for variant characteristics (e.g., infectiousness, severity of illness, mechanism of immune evasion) are difficult to capture.”

Response to Reviewer 3:

General comment

Thank you for these revisions and explanations. I am largely satisfied with these changes, with the following remaining points (apologies, the first two are new).

Response: Thank you for these additional comments, which we address below.

Comment 1

In the text I see the term “uncertainty interval”, but would specify that these are 95% intervals, if that is the case (also in table footnotes). And in methods could note whether these are equal-tailed or HPD intervals. Given the question below, the authors might also describe somewhere how the point estimates are calculated (mean of simulations, median of simulations, model evaluated using parameter means, etc), if not done already.

Response: We appreciate the opportunity to clarify the interval. This is an uncertainty interval that represents the entire range (100%) of the varied parameters. This interval is generated by simulating a full range of multiple model inputs at baseline (as defined below), then running the model; we report the full range of values as the bounds of the interval. The point estimate is from the base case assumption of model inputs (not mean or median); therefore, the uncertainty interval bounds are expected to be asymmetric relative to the point estimate. We have further clarified these points in the Methods section and the Table footnotes and Figure legends.

In Methods:

“We generated uncertainty intervals for the primary analysis based on parameter uncertainty in vaccine effectiveness and waning over time, baseline seroprevalence levels, and non-severe infection multipliers (see Appendix; **Table A5**). **This interval is generated by simulating the full range of multiple model inputs at baseline, which define the bounds of the interval; the reported point estimate uses the base case assumption of model inputs, so the bounds are expected to be asymmetric relative to the point estimate.**”

In Table/Figure Legend:

Table 1: Number of severe COVID-19 cases, risk, and number needed to treat to avert severe COVID-19 in six risk groups with different frequencies of COVID-19 booster vaccination.

In footnote: **The uncertainty intervals and capture the full range of varied model parameters, while the point estimate uses base case assumptions of model inputs.**

“Figure 1: Sensitivity analysis of model parameters for COVID-19 risk and booster vaccination... **The vertical bars represent uncertainty intervals and capture the full range of varied model parameters, while the point estimate uses base case assumptions of model inputs.**”

We made this revision for Figure 2 and 3 as well.

Comment 2

Though I am not in a position to confirm whether this is actually an issue, the comparison of point estimates and interval bounds in Figure 2 B & C is surprising to me. In these plots, the point estimate appears much closer to the lower bound than the upper bound (in some cases, point estimate minus lower bound appears $\frac{1}{4}$ the value of upper bound minus point estimate). In Figure 3 the point estimates are more modestly shifted towards the top of the interval, so I suspect this is not an inherent feature of the quantities being modelled. Is the point estimate calculated as the mean or median of the simulation results used to construct the interval? If not (e.g., if calculated from parameter point estimates) that could potentially explain this issue. The interval could also look strange if constructed from a very small number of parameter sets. Anyway, I would suggest tracking down why this is happening to confirm that there is no problem with the intervals or point estimates presented.

Response: We appreciate the reviewer’s attention here. As discussed in Comment #1, the point estimate is from the base case assumption of model inputs (not mean or median of uncertainty analysis). The uncertainty interval represents the full range of values based on simulating a full range of multiple model inputs at baseline. Therefore, the uncertainty interval bounds are expected to be asymmetric relative to the point estimate. Furthermore, Figure 2 and 3 are different models, as Figure 2 uses the primary model, while Figure 3 uses the dynamic transmission model. We have further clarified these points in the Methods section and the Table footnotes and Figure legends (as described in Comment #1).

See Comment #1 for changes to Methods, Figure legends, and Table footnotes

Comment 3

For the comment from the first review on the use of an NNT of 48: in the revision the number 48 is removed, but the text still states “common primary care measures have a range of NNT from 50-400”. If the NNT of 48 for influenza is what justifies the 50 in “50-400”, then I think the issue still remains.

Response: We have revised this as shown below.

In Discussion:

“For example, common primary care measures have a range of NNT often below 1000, such as...”

REVIEWERS' COMMENTS

Reviewer #2 (Remarks to the Author):

I thank the authors for once again responding to my comments. I am now happy that all my concerns have been addressed and can support publication.